# Single-cell RNA sequencing reveals the cellular and molecular heterogeneity of treatment-naïve primary osteosarcoma in dogs
Dylan T. Ammons [1] ✉, Leone S. Hopkins[2], Kathryn E. Cronise[1], Jade Kurihara[2], Daniel P. Regan[1,2] & Steven Dow [1,2] ✉

Osteosarcoma (OS) is a heterogeneous, aggressive malignancy of the bone that disproportionally affects children and adolescents. Therapeutic interventions for OS are limited, which is in part due to the complex tumor microenvironment (TME). As such, we used single-cell RNA sequencing (scRNA-seq) to describe the cellular and molecular composition of the TME in 6 treatment-naïve dogs with spontaneously occurring primary OS. Through analysis of 35,310 cells, we identified 41 transcriptomically distinct cell types including the characterization of follicular helper T cells, mature regulatory dendritic cells (mregDCs), and 8 tumor-associated macrophage (TAM) populations. Cell-cell interaction analysis predicted that mregDCs and TAMs play key roles in modulating T cell mediated immunity. Furthermore, we completed cross-species cell type gene signature homology analysis and found a high degree of similarity between human and canine OS. The data presented here act as a roadmap of canine OS which can be applied to advance translational immuno-oncology research.

Osteosarcoma (OS) is an aggressive malignancy of the bone that disproportionally impacts children and adolescences. Despite a profound impact on the lives of affected individuals, effective therapeutics are lacking, with minimal advancements since the introduction of combined surgical excision and adjuvant chemotherapy in 1986[1]. Slow advancements in the development of OS therapeutics are, in part, due to the relatively rare incidence which limits patient accrual into clinical trials. In recent years, there has been increased interest in using large animal models to evaluate and validate the potential of immunotherapeutics for various cancers[2,3]. Spontaneously occurring canine OS is regarded as an ideal model of human OS due to higher disease prevalence in dogs, similar genetics and pathology, and the immune competent status of dogs[4]. Although dogs have been identified as a valuable pre-clinical model, species-specific reagent limitations have restricted researcher's ability to fully characterize the canine OS tumor microenvironment (TME).

Osteosarcoma has a complex TME that consists of malignant osteoblasts, osteoclasts, fibroblasts, macrophages, and lymphocytes as well as numerous other stromal and immune components. Together, the OS TME creates an immune suppressive milieu that hinders antitumor immune responses[5]. Researchers have turned to the TME with the objective of understanding and targeting the cellular constituents that promote immune suppression[6,7]. Unlike many other cancer types, there have been reports in both humans and dogs that suggest increased macrophage abundance in OS reduces metastatic rate and enhances survival[8–10]. This unexpected finding and the ill-defined mechanisms of immune suppression in the OS TME highlights the need for a deeper understanding of OS pathobiology.

In recent years, single-cell RNA sequencing (scRNA-seq) has emerged as a valuable tool to investigate the transcriptomes of individual cells within heterogenous tissues. The approach overcomes species-specific regent limitations by relying on a universal transcript capture method that is only limited by the completeness of genome annotations[11]. Importantly, the human scRNA-seq landscapes of primary, recurrent, and metastatic OS have recently been described and act as a point of reference for cell type homology analysis between canine and human OS[12,13]. The aim of the

[1]Department of Microbiology, Immunology, and Pathology, College of Veterinary Medicine and Biomedical Sciences, Colorado State University, Fort Collins, CO, USA. [2]Flint Animal Cancer Center, Department of Clinical Sciences, College of Veterinary Medicine and Biomedical Sciences, Colorado State University, Fort Collins, CO, USA. ✉e-mail: dylan.ammons@colostate.edu; steven.dow@colostate.edu

current study was to use scRNA-seq to complete a molecular dissection of the canine OS TME and evaluate cell type transcriptomic homologies between humans and dogs.

In the present study, we generated a single-cell RNA sequencing reference dataset of six treatment-naïve dogs with primary osteosarcoma. Our analysis revealed the presence of 41 transcriptomically distinct cell types in canine OS and provided evidence of conserved cell type gene signatures between human and canine OS. Overall, the data generated here can be used to inform the identification of conserved OS TME features and facilitate further study of the canine osteosarcoma tumor microenvironment.

## Results

### Establishment of a treatment-naïve canine osteosarcoma reference database

To establish a treatment-naïve canine osteosarcoma reference, we completed single-cell RNA sequencing (scRNA-seq) on 6 dogs and collected data on a total of 35,310 cells. The average number of cells collected per tumor was 5885 and on average each cell was sequenced to a depth of 72,649 reads per cell (Supplementary Table 3). All tumors were confirmed to be osteosarcoma by histology and histological subtyping was completed on each tumor (Table 1). In total, the curated dataset consisted of 1 fibroblastic, 1 chondroblast, and 4 osteoblastic tumors, with one dog exhibiting radiographical evidence of lung metastasis.

Initial low-resolution cell type annotation revealed the presence of 7 major cell types consisting of T cells, B cells, tumor infiltrating monocytes (TIMs)/tumor associated macrophage (TAMs), dendritic cells (DCs), osteoclasts (OCs), tumor cells, cycling tumor cells, and an additional 3 minor cell populations consisting of neutrophils, mast cells, and endothelial cells (Fig. 1a). Evaluation of the dataset for evidence of batch effects indicated uniform distribution of cell types between biological replicates (Fig. 1b). The one exception was that naïve dog 6 had a higher proportion of neutrophils compared to the other study dogs. This skew might be a result of sampling bias in which necrotic tumor, blood, or bone marrow contamination was introduced during sampling. Subsequent analysis of cell type proportions revealed 42.3% of the dataset consisted of tumor cells or fibroblasts, 2.1% of the dataset was endothelial cells, and the remaining 55.6% was composed of immune cells (Fig. 1c).

Cell types were annotated based on expression of canonical markers, reference mapping using a human OS dataset, and gene set enrichment analysis (Fig. 1d). Cell type gene lists used by Liu et. al. to define cell populations in human OS were applied using module scoring to provide further support for cell classifications (Supplementary Fig. 1a)[12]. These approaches consistently enabled the identification of T cells, B cells, osteoclasts, and endothelial cells. However, our high-level unsupervised clustering failed to distinguish between stromal fibroblasts and malignant osteoblasts. This unexpected observation may be in part due to the presence of a fibroblastic osteosarcoma tumor in our dataset and the broad expression of fibroblast markers (FAP, FBLN1) across all tumor cell clusters (Supplementary Fig. 1b).

Due to the inability to identify a distinct fibroblast population using feature expression, we applied CopyKAT and inferCNV to complete copy number variation (CNV) analysis to infer which cells exhibited aneuploidy based on their global transcriptional properties (Fig. 1e)[14,15]. The analysis

revealed that the majority of cells in the tumor/fibroblast cluster exhibited evidence of CNV aberrations with only a small subset of cells predicted to be diploid (Fig. 1e; purple arrow, Supplementary Figs. 2–9). The diploid cells were determined to represent a small cluster of fibroblasts which were investigated further through subclustering analysis.

### Dissection of the tumor and stromal populations reveals a distinct fibroblast cluster

Subclustering analysis on cycling tumor cells and tumor/fibroblasts identified 10 distinct cell clusters which we defined as 4 cycling malignant osteoblasts clusters, 5 non-cycling malignant osteoblast clusters, and 1 fibroblast cluster (Fig. 2a, Supplementary Fig. 10a). The defining features for each cluster were identified using a Wilcoxon Rank Sum test and the top 3–5 unique features were visualized using a heatmap and feature plots (Fig. 2b, c). Overall, the malignant osteoblasts exhibited a unique gene expression profile with collagen genes and alkaline phosphatase (ALPL) contributing to the gene signatures. We observed a small cluster of tumor cells (c9) that exhibited a gene expression pattern (OAS1, ISG15, OAS2) consistent with an interferon (IFN) response gene signature (Fig. 2b). This observation was further supported through completion of GSEA using Hallmarks gene set terms (Fig. 2d). Similar IFN signature enriched cells have been reported among immune cells, but the observation of such a cluster in a tumor population has not been previously reported in human OS studies[12,13,16,17]. Interpretation of GSEA further revealed that fibroblasts (c6) exhibited the most pronounced "epithelial-mesenchymal transition" (EMT) and "angiogenesis" signatures, which suggests the fibroblasts might play a role in promoting tumor growth. Additionally, GSEA supported the annotation of hypoxic osteoblasts (c4), as the cluster exhibited the strongest hypoxic transcriptomic signature.

To confirm the identification of fibroblasts, we used module scoring with a human fibroblast gene list[18]. This analysis confirmed Cluster 6 exhibited the strongest fibroblast gene signature (Supplementary Fig. 10b). We then completed differential gene expression (DGE) analysis contrasting fibroblasts (c6) and non-hypoxic osteoblasts (c0, c1, and c2) to better define the canine fibroblast gene signature (Fig. 2e; Supplementary Data 3). While key fibroblast markers such as FAP and ACTA2 were identified, the top features consisted of SFRP2 and PRSS23 which were recently reported to be associated with a fibroblast population involved in wound healing[19]. To conclude our analysis of tumor cells, we sought to further investigate the transcriptomic signature of hypoxic osteoblasts (c4) by contrasting with non-hypoxic osteoblasts (c0, c1, and c2). Few differentially expressed genes were identified, suggesting the cell types are similar, but subsequent pathway analysis identified enrichment of "response to oxygen levels" to be a top enriched pathway, suggesting that the tumor cells were indeed hypoxic (Fig. 2f, Supplementary Data 3, Supplementary Fig. 10c). In summary, we were able to resolve a population of fibroblasts through completion of subclustering analysis, as well as define the transcriptional heterogeneity within malignant osteoblasts.

### Subclustering analysis reveals a population of CXCL13+ follicular helper CD4 T cells

To ensure we captured all biologically relevant T cell populations, we completed subclustering analysis, which led to the identification of 10

## Table 1 | Study dog demographics

| Dog ID | Sex | Breed | Age (years) | Tumor location | Evidence of metastasis | Histological subtype |
|--------|-----|-------|-------------|----------------|------------------------|----------------------|
| Naïve 1 | FS | Mixed (Husky) | 8 | L proximal humerus | No | Osteoblastic |
| Naïve 2 | MC | Catahoula | 11.5 | R distal femur | Yes | Osteoblastic |
| Naïve 3 | MC | Labrador Retriever | 7.8 | L distal femur | No | Fibroblastic |
| Naïve 4 | MC | Great Dane | 8 | R distal radius | No | Osteoblastic |
| Naïve 5 | FS | Mixed | 11.3 | R distal radius | No | Chondroblastic |
| Naïve 6 | FS | Catahoula | 8.4 | R distal radius | No | Osteoblastic |

transcriptomically distinct clusters: 3 CD8 T cell, 4 CD4 T cell, 1 NK cell, and 2 mixed CD4/CD8 T cell clusters (Fig. 3a, b). Next, we interrogated T cell subtypes using an approach that has been applied in human breast cancer and OS scRNA-seq datasets to describe T cell populations[13,20]. We modified

the gene lists used in previous applications to include signatures for cycling T cells, NK cells, and IFN-signature T cells that we recently established in circulating canine leukocytes[21]. Overall, this approach proved to be consistent with annotations assigned using canonical markers (Fig. 3c).

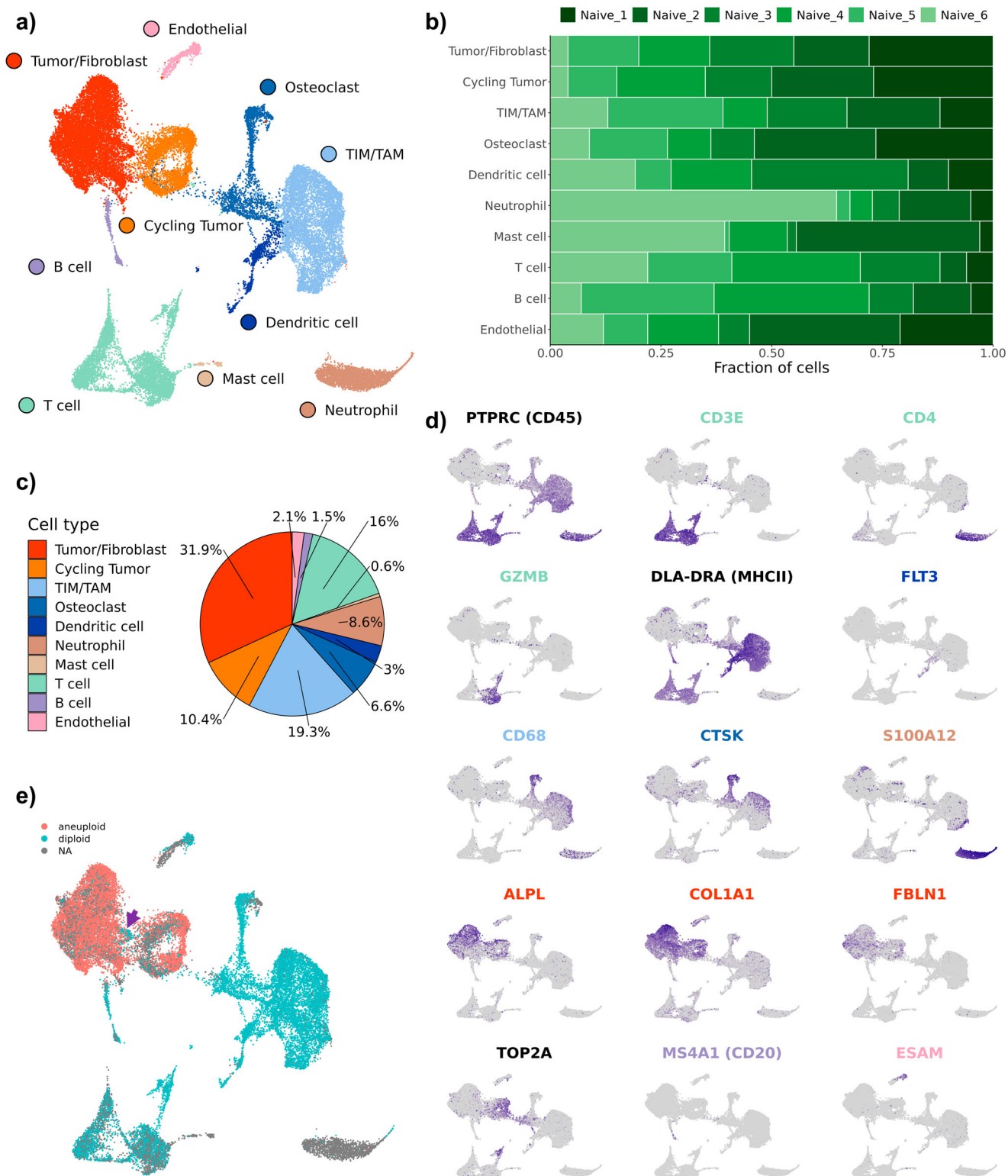

**Fig. 1 | Unsupervised clustering reveals 10 distinct cell types in canine osteosarcoma. a** UMAP representation of 35,310 cells obtained from the primary tumors of 6 dogs diagnosed with osteosarcoma. **b** Stacked bar graph depicting the cell type proportion contributed by each dog. **c** Pie chart depicting the cellular composition of the data as a percentage of total cells. **d** Feature plots depicting the log normalized

counts of canonical markers used to justify major cell type classifications. **e** UMAP depicting the results of CopyKAT copy number variation prediction. Gray (NA) values indicate that the cell did not have a large enough transcriptome to be evaluated using CopyKAT. The purple arrow points to the identified fibroblast population.

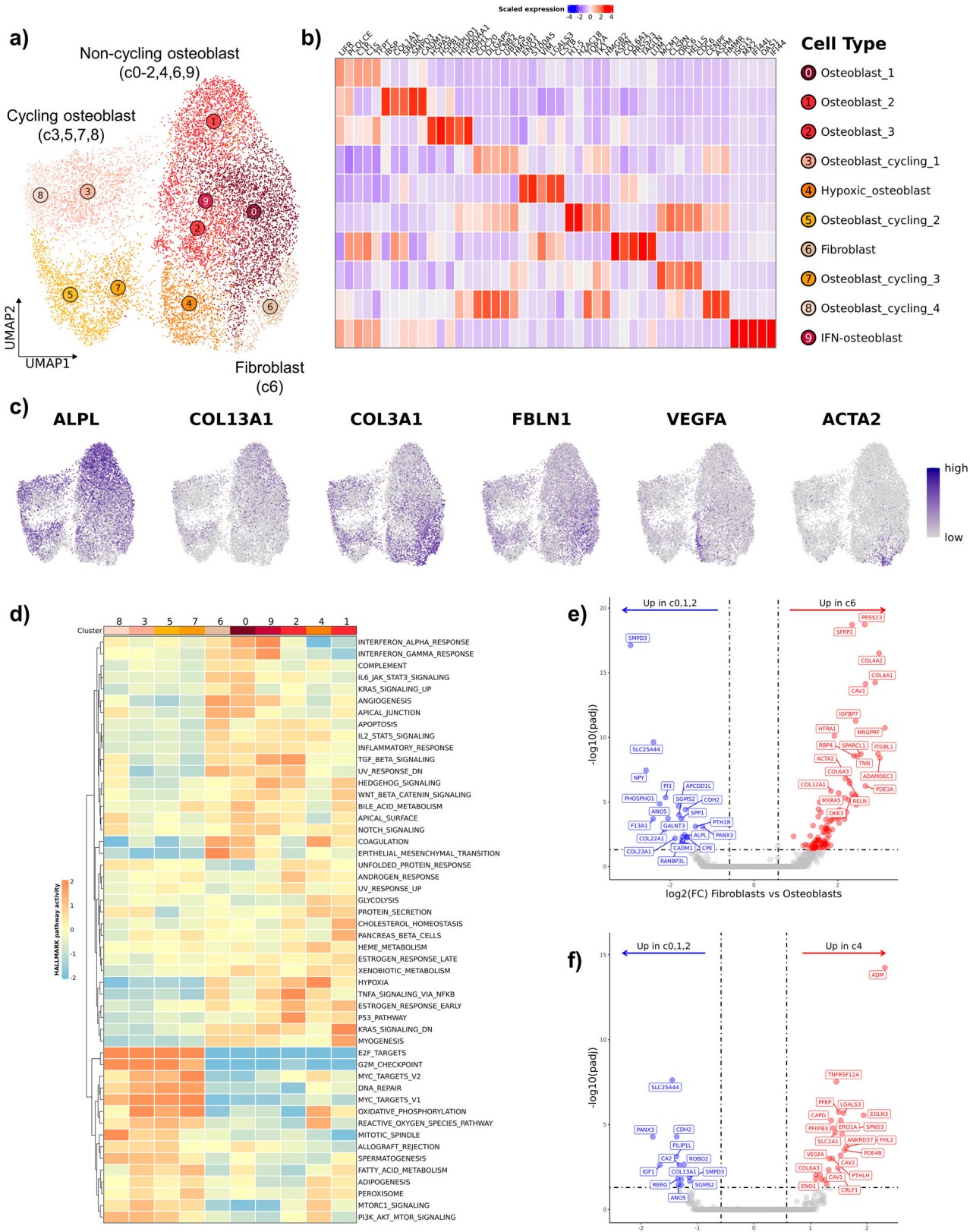

**Fig. 2 | Subclustering analysis of tumor and stomal cells reveals osteoblast heterogeneity and a distinct fibroblast cluster. a** UMAP representation of tumor and stomal cells (*n* = 17,904 cells) depicting the 10 clusters identified though unsupervised clustering. **b** Heatmap depicting expression of the top 3–5 features that define each cluster. **c** Feature plots illustrating log normalized counts. **d** Heatmap of scaled gene set enrichment analysis pathway activity for HALLMARK terms, with

hierarchical clustering of rows and columns. Dendrogram of terms using Euclidean distance shown on the left. Volcano plots depicting the results of pseudobulk differential gene expression analysis for (**e**) fibroblasts (c6) versus osteoblasts (c0, c1, c2) and (**f**) hypoxic osteoblasts (c4) versus non-hypoxic osteoblasts (c0, c1, c2). The top 20 features (weighted by adjusted *P* value) are labeled.

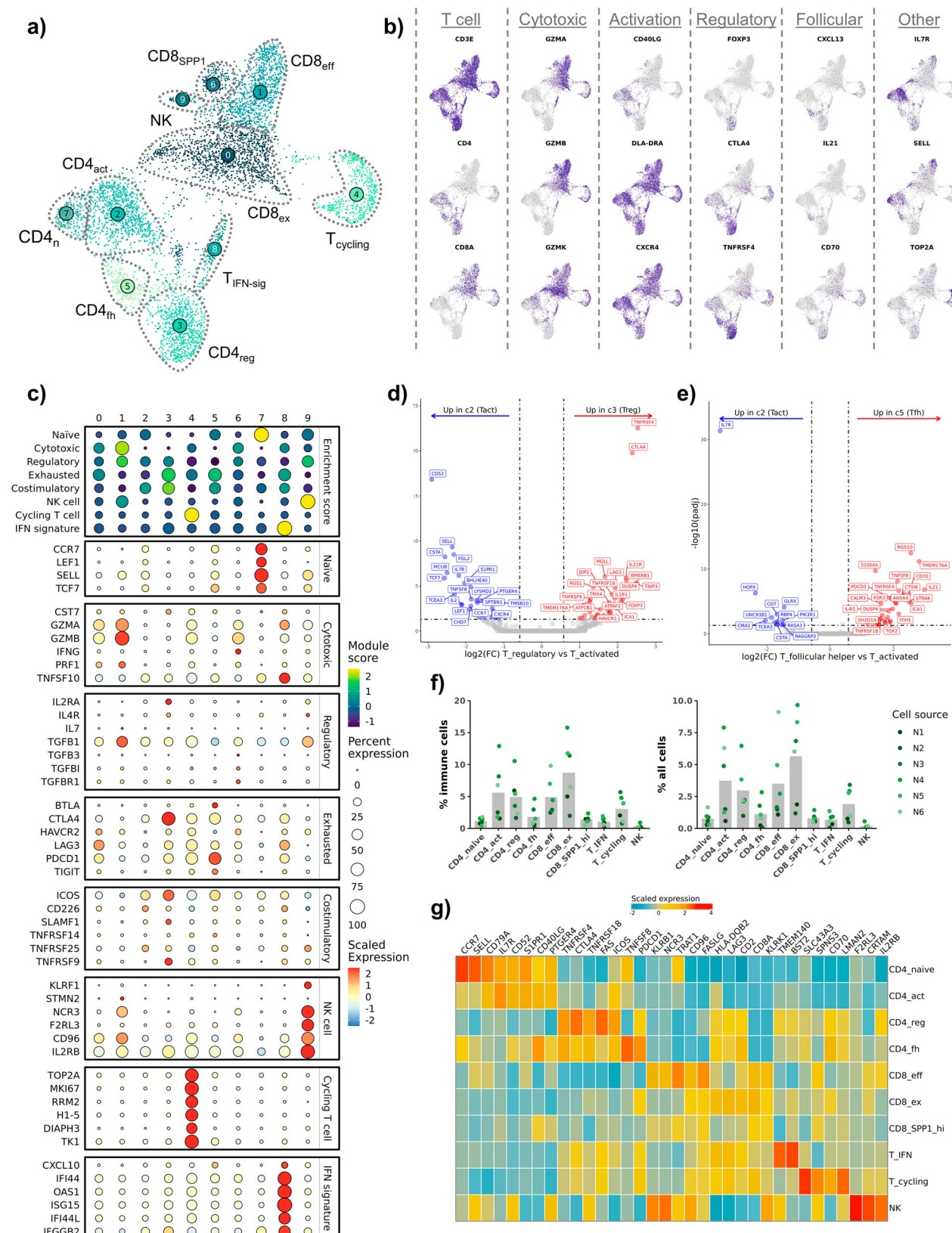

Although the gene signatures were definitive for naïve CD4 T cells and cytotoxic CD8 T cells, other gene signature scores provided weaker support for their corresponding cell types. For instance, regulatory T cells (T$_{regs}$) and follicular helper CD4 T cells (CD4$_{fh}$) both exhibited moderate enrichment for exhausted and costimulatory terms, with minimal distinction between the two T cell types. The analysis also revealed the presence of a T cell cluster with an IFN gene signature, a population that has been reported to be hypersensitive to stimulation[17].

After identifying each T cell subset, we completed pseudobulk conversion and DGE analysis to further establish the transcriptomic signatures

**Fig. 3 | Analysis of tumor infiltrating NK and T cells provides gene signatures for regulatory and follicular helper T cells. a** UMAP representation of NK, CD4, and CD8 T cells (n = 5778 cells) colorized by cell subtypes. **b** Feature plots depicting expression of canonical T cell markers. **c** Dot plots depicting the scaled module score ("Enrichment score" facet) and the scaled expression of features associated with each of the 8 gene lists used to calculate the enrichment scores. Volcano plots depicting the results of pseudobulk differential gene expression analysis for **(d)** regulatory T cells (c3) versus activated CD4 T cells (c2) and **(e)** CD4 follicular helper T cells (c7) versus activated CD4 T cells (c2). The top 20 features (weighted by adjusted P value) are labeled

for each plot. **f** Bar chart depicting mean composition of each cluster as a percentage of immune cells and percentage of total cells (osteoclasts were included as an immune cell in the calculation). Each dot corresponds to a biological replicate. **g** Scaled expression of the top 3–5 features for each cluster. The selected features were chosen based on the results of FindAllMarkers() and the inclusion of the feature in the surfaceome database listed as predicted to have surface expression. Abbreviations: $CD8_{SPP1}$ = SPP1$^+$ CD8 T cell, $CD4_{act}$ = activated CD4 T cell, $CD4_n$ = naïve CD4 T cell, $CD4_{fh}$ = follicular helper T cell, $CD4_{reg}$ = regulatory T cell, $T_{IFN-sig}$ = T cells enriched in interferon gene signatures, NK = natural killer cell, $CD8_{eff}$ = effector CD8 T cell, $CD8_{ex}$ = exhausted CD8 T cell.

of $T_{regs}$ and $CD4_{fh}$. Comparisons between $T_{regs}$ (c3) and activated CD4 T cells (c2) revealed overexpression of *IL21R*, *TNFRSF4*, and *TNFRSF18*, with *CTLA4* being the most definitive marker of $T_{regs}$ (Fig. 3d, Supplementary Data 3). When repeating this analysis on $CD4_{fh}$ (c5) cells, we identified *CXCL13*, *IL4I1*, and *TMEM176A* to be defining features (Fig. 3e, Supplementary Data 3). Although intratumoral $CD4_{fh}$ cells exhibited a distinct exhaustion profile (*PDCD1*, *TOX*, *TOX2*, *IL4I1*), they also displayed a gene signature consistent with follicular helper T cells (*CXCL13*, *IL21*, *CD70*)[22,23]. A similar population of *CXCL13*$^+$ $CD4_{fh}$ T cells has been identified in multiple human tumors and the cell type has been implicated in tertiary lymphoid follicle formation and modification of intratumoral adaptive immune responses[24–26]. Our analysis confirms expression of *CTLA4/FOXP3* on $T_{regs}$ and *CXCL13/IL21* on $CD4_{fh}$ is conserved across species, while also providing complete gene signatures for the canine T cell subtypes[18].

Following initial cell classification, we determined the cellular composition of each cell type as a percentage of immune cells and of all cells within each sample (Fig. 3f, Supplementary Table 4). This analysis revealed exhausted CD8 T cells ($CD8_{ex}$) and effector CD8 T cells ($CD8_{eff}$) to be among the most abundant populations, along with activated CD4 T cells and $T_{regs}$. We then curated a heatmap of defining features predicted to be expressed on the cell surface with the objective of identifying potential cell markers to be used in alternative cell identification approaches, such as flow cytometry (Fig. 3g, Supplementary Fig. 11, Supplementary Data 4)[27]. With the caveat that transcript presence does not always correlate with protein expression, the analysis suggested that *TNFRSF4* (OX-40), *TNFSF8* (CD153), and *TMEM140* may represent valuable surface markers for further investigation of canine $T_{regs}$, $CD4_{fh}$, and $T_{IFN-sig}$, respectively. Together, the relative cellular percentages and potential surface markers provide a foundation for further functional study of the cell types identified in our transcriptomic analysis.

## Mature regulatory dendritic cells are present in canine OS and are predicted to modulate T cell mediated immunity

Five dendritic cell (DC) subtypes were identified when completing subclustering analysis on FLT3$^+$ cells. The subtypes identified included conventional DC2s (cDC2; c0), mature regulatory DCs (c1; mregDC), cDC1s (c2), plasmacytoid DCs (c3; pDC), and precursor DCs (c4; preDC) (Fig. 4a). Key features used to assign cell type identities included *DNASE1L3* (cDC1), *CCR7/IL4I1* (mregDCs), *CD1C* (cDC2), *IL3RA* (preDC), and *IGKC* (pDC) (Fig. 4b)[28,29]. The population of canine preDCs closely resembled a recently redefined human preDC cell type that exhibits a tendency to cluster with pDCs when investigated using scRNA-seq[28]. Of note, we previously identified cDC2, cDC1, preDC, and pDC cell types in canine peripheral blood, however mregDCs (c1) were not observed, suggesting a potential tissue specificity[21]. The identification of mregDCs, also reported as migratory (mig) DCs, is of note as this cell type is predicted to modulate T cell responses[30,31]. Thus, we provide evidence that a key cell type reported to have immune regulatory properties is present in canine OS.

We next used hierarchical clustering and Toll-like receptor expression to investigate differences between preDCs and pDCs. Hierarchical clustering indicated preDCs are closely related to myeloid cDC2s and cDC1s, while pDCs were located on a unique clade (Fig. 4c). In humans, pDCs are reported to exhibit high expression of *TLR9* and *TLR7*, which we identified

to be highly expressed on canine pDCs (Fig. 4d)[32]. To ensure that none of the DC populations were of B cell origin, we evaluated *MS4A1* (CD20) expression and found it to be minimally expressed (Supplementary Fig. 12a). We then used pySCENIC to predict active regulons in each DC subtype (Supplementary Fig. 12b). This analysis revealed *TCF4* and *RUNX2*, master regulators of pDC development, to be enriched in both pDCs and preDCs[33]. Overall, these findings suggest canine preDCs are closely related to the recently defined plasmacytoid-like human preDCs[28].

To confirm mregDCs exhibited a mature, immune regulatory transcriptomic signature, we used module scoring with gene lists previously applied to investigate human DC subtypes[30,34]. This analysis revealed that mregDCs had a marked enrichment for migration, regulatory, and maturation associated gene signatures (Fig. 4e). Subsequent, DGE analysis of canine mregDCs (c1) relative to cDC2s (c0) revealed a distinct mregDC signature of *CCR7*, *IL4I1*, *CCL19*, and *FSCN1* with substantial overlap to the human mregDC transcriptional program (Fig. 4f, Supplementary Data 3)[30]. With the precedent that mregDCs interact with intratumoral T cells to shape adaptive immune responses in humans, we wanted to determine whether a similar interaction might occur between mregDC and T cells in dogs[24,25]. We used CellChat to evaluate interactions between mregDCs and T/NK cells[35]. This analysis revealed enriched PD-1/PD-L1 and CTLA4/CD80 interactions between mregDCs and CD4 $T_{regs}$, $T_{fh}$ cells, and naïve T cells (Fig. 4g). In summary, we present the transcriptomic signature of canine mregDCs and provide evidence of intratumoral interactions between canine mregDCs and T cells.

## Macrophage transcriptomic states support a spectrum of cell types

Due to the transcriptional overlap between tumor associated macrophages (TAMs) and osteoclasts (OCs), we analyzed these two cell types in the same UMAP space. In doing so, our analysis highlighted the relatedness of OCs and TAMs which would have been overlooked if analyzed independently. Through subclustering analysis we identified 8 transcriptomically distinct macrophage/monocyte populations which were annotated using modified nomenclature derived from Ma et al. (Fig. 5a–c)[36]. Activated TAMs (c0, TAM_ACT) and intermediate TAMs (c1, TAM_INT) did not fit into any of the macrophage subtypes presented in Ma et al. so they were instead annotated based on an activated signature (*CD5L*, *CD40*, *CD80*) and an intermediate polarization signature, respectively. Tumor infiltrating monocytes (TIMs) were divided into two populations based on CD4 expression; a division of monocytes unique to dogs[21,37]. Unsupervised clustering divided lipid-associated (LA-) TAMs into two subclusters defined by either *C1QC*$^{hi}$ expression (c3) or *SPP2*$^{hi}$ expression (c2). To better define the distinctions between the two LA-TAM populations we completed pseudobulk-based DGE analysis (Fig. 5d, Supplementary Data 3). The analysis revealed *IL2RA*, *CXCL10*, and *SERPING1* as key markers of *C1QC*$^{hi}$ LA-TAMs, while *ENO1*, *LGALS3*, and *RBP4* defined *SPP2*$^{hi}$ LA-TAMs. Based on the analysis, *C1QC*$^{hi}$ LA-TAMs appear to most closely resemble the definitions of human LA-TAMs provided by Ma et al.

In addition to the recently proposed TAM nomenclature, we used module scoring with pro- and anti-inflammatory gene lists to investigate the macrophage populations in a more traditional dichotomy (Supplementary Table 5)[38]. We identified the *C1QC*$^{hi}$ LA-TAM (c3) cluster to have the strongest anti-inflammatory transcriptomic signature while CD4$^+$

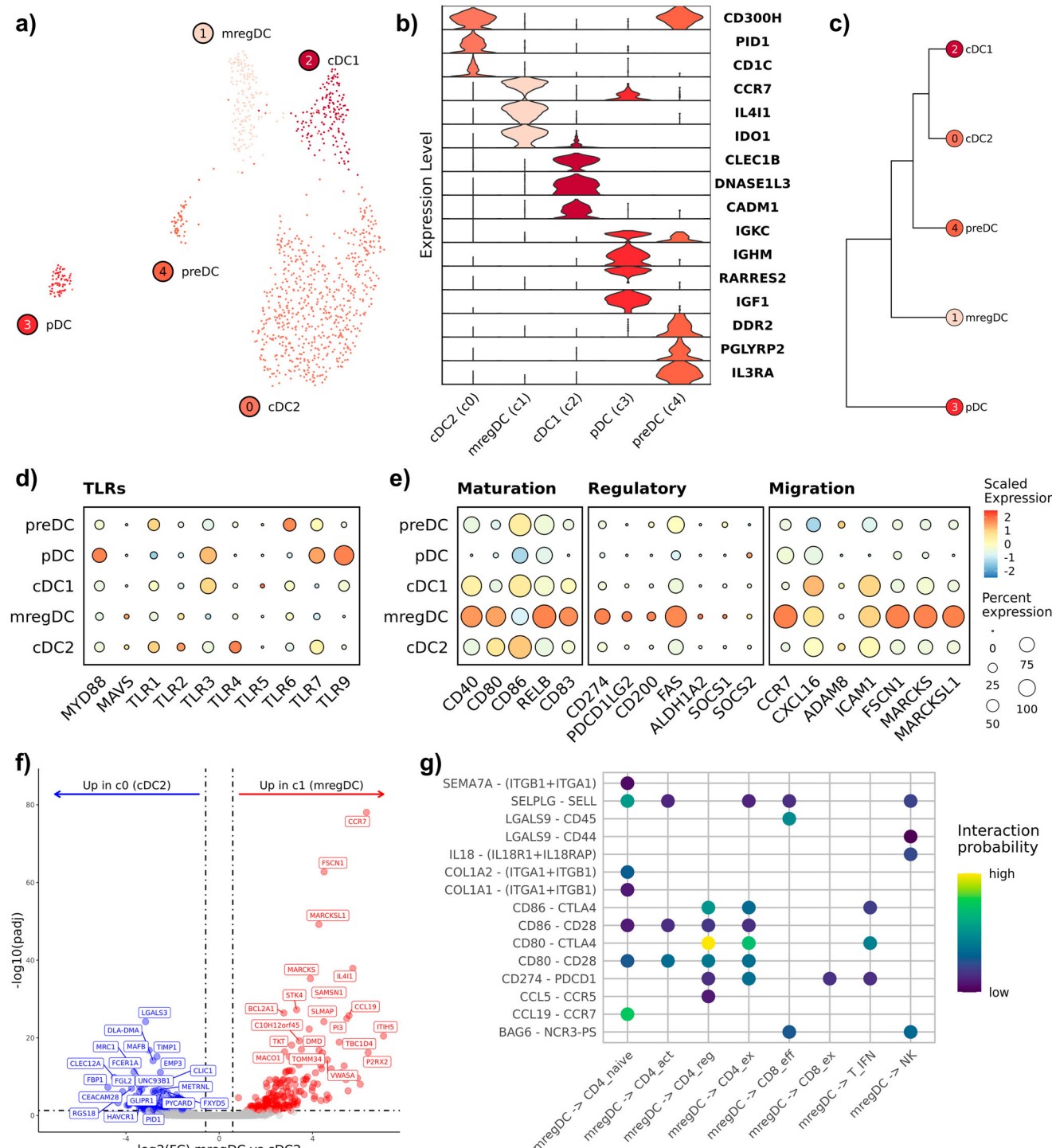

**Fig. 4 | Mature regulatory dendritic cells (mregDCs) are present in canine OS tissues and are predicted to interact with T cells. a** UMAP representation of dendritic cells (DCs) colorized by cell subtype (*n* = 1067). **b** Violin plots depicting expression of key DC features used for cell identification. **c** Dendrogram depicting results of hierarchical clustering of DC subtypes using cluster averaged log normalized expression. **d**, **e** Dot plots depicting scaled expression of Toll-like receptors (TLRs), DC maturation, regulatory, and migratory features. **f** Volcano plots depicting the results of pseudobulk differential gene expression analysis for mregDCs (c1) versus cDC2s (c0). The top 20 features (weighted by adjusted *P* value) are labeled for each plot. **g** Dot plot depicting the interaction probability (as determined using CellChat) of signaling networks for each mregDC-T cell interaction. mregDC mature regulatory DC, cDC2 conventional DC2, cDC1 conventional DC1, pDC plasmacytoid DC, preDC precursor DC.

monocytes (c11) exhibited the most prominent pro-inflammatory transcriptomic signature (Fig. 5e). To further investigate the gene signatures of Clusters 11 and 3 we completed pseudobulk-based DGE analysis (Fig. 5f, Supplementary Data 3). The genes upregulated in Cluster 11 exhibited overlap with the predefined gene set that was used to identify the cluster as

pro-inflammatory, while also revealing *IL1B*, *S100A12*, *LTF*, and *VCAN* as defining features. DGE analysis of the anti-inflammatory cluster (c3) exhibited less overlap with the gene list originally used to identify the cluster as anti-inflammatory (with *MRC1* the only overlapping feature), but the analysis further revealed *APOE*, *IGF1*, and complement receptors *C1QA/B/C*

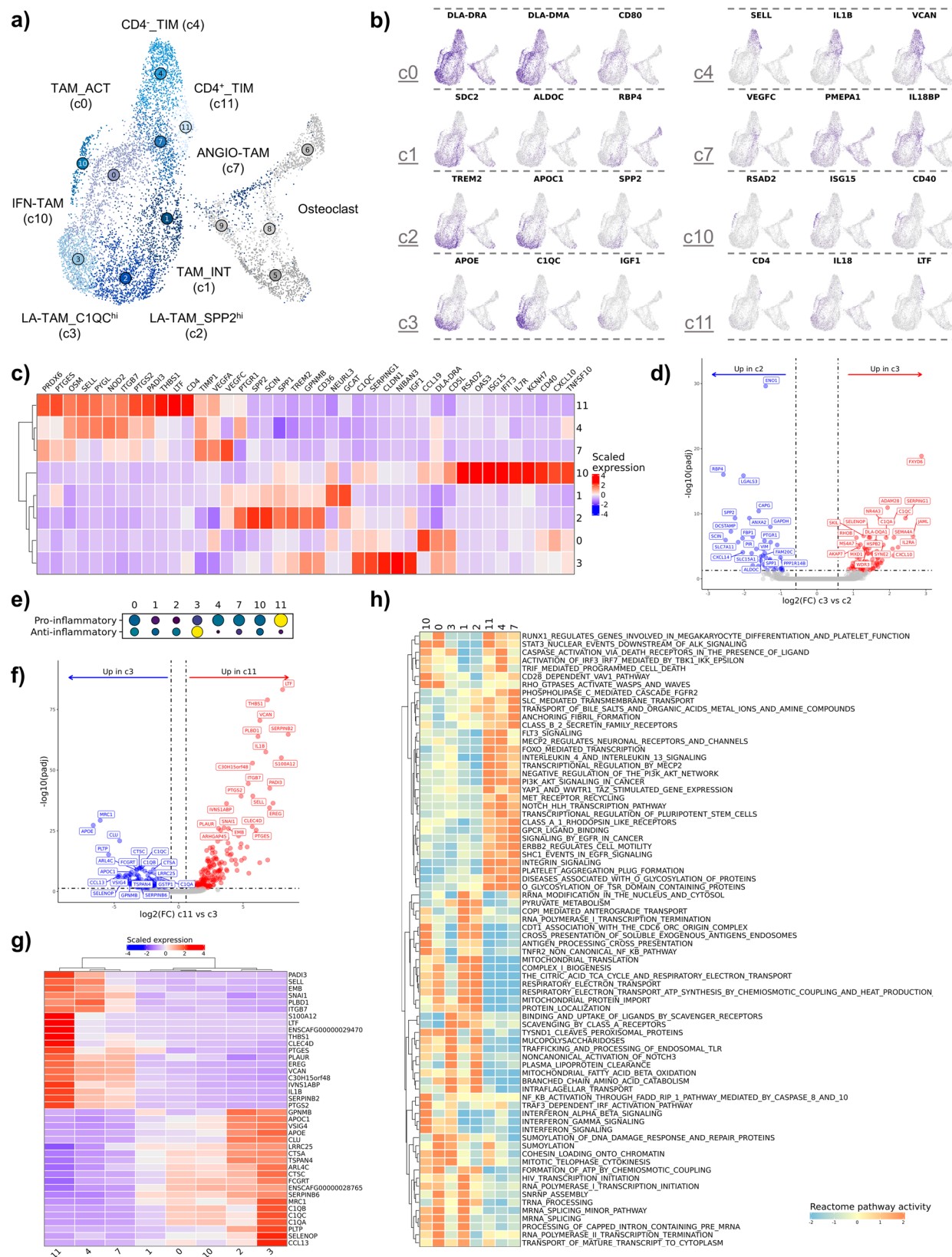

as enriched markers. The top features identified when contrasting Clusters 11 and 3 were then used to generate a heatmap to evaluate how the expression of these features varied across all macrophage clusters (Fig. 5g). Findings from the analysis suggested that there is a spectrum of macrophage phenotypes, which is consistent with human macrophage literature[39]. As such, we next

sought to better define the heterogeneity of the macrophage populations without relying on predefined cell type gene signatures.

Gene set enrichment analysis was used to provide further insights into the inferred functional capacity of each macrophage subtype (Fig. 5h). Cell clusters 4, 7, and 11 clustered together based on pathway enrichment scores

**Fig. 5 | Tumor-associated macrophages exhibit a spectrum of transcriptional states with C1QC[hi] LA-TAMs exhibiting the most immune suppressive transcriptional profile. a** UMAP representation of tumor infiltrating monocyte (TIM), tumor associated macrophage (TAM), and osteoclast (OC) populations ($n = 10,512$). Four osteoclast (OC) clusters are depicted but grayed out. **b, c** Feature plots and heatmap of canonical features used to define cell types. **d** Volcano plots depicting the results of pseudobulk differential gene expression analysis for C1QC[hi] lipid-associated (LA)-TAMs (c3) versus SPP2[hi] LA-TAMs (c2). The top 20 features (weighted by adjusted $P$ value) are labeled for each plot. **e** Dot plot depicting module scoring of "Pro-inflammatory" and "Anti-inflammatory" gene signatures. The size of each dot indicates the percentage of cells enriched for a given gene set (larger dot equates to more broadly enriched), and the color indicates strength of enrichment score (brighter color indicates stronger score). **f** Volcano plot depicting results of differential gene expression analysis when contrasting the cluster with the most pro-inflammatory gene signature (c11; CD4[+] TIMs) to the cluster with the most anti-inflammatory gene signature (c3; C1QC[hi] LA-TAMs). The top 20 features (weighted by adjusted $P$ value) are labeled for each plot. **g** Heatmap of the top 20 features identified to be upregulated in in c11 relative to c3 and vice versa. Columns are ordered by hierarchical clustering (Euclidean distance), shown at top. **h** Heatmap of scaled gene set enrichment analysis pathway activity for Reactome terms, with hierarchical clustering of rows and columns. Dendrogram of terms using Euclidean distance shown on the left. TAM_ACT activated TAM, ANGIO-TAM pro-angiogenesis TAM, TAM_INT intermediate TAM, IFN-TAM TAM enriched in interferon gene signatures.

suggesting the three transcriptionally distinct clusters have similar underlying gene signatures. LA-TAMs (c2, c3) and intermediate TAMs (c1) exhibited the strongest scavenger receptor associated activation, suggesting a mature macrophage population with immune suppressive properties[40]. Several terms were identified suggesting that both SPP2[hi] LA-TAMS and intermediate TAMs preferentially utilize oxidative phosphorylation and mitochondrial metabolic pathways. C1QC[hi] LA-TAMs had a distinct profile suggestive of lipid and polysaccharide metabolism. Lastly, GSEA confirmed c10 to be consistent with IFN-TAMs based on strong enrichment of IFN signaling associated terms. In summary, we described the transcriptional profiles of macrophages in the canine OS TME which provides a foundation for further investigation of the functional relevance of each cell type.

### Analysis of canine osteoclasts reveals four transcriptomically distinct populations

Within the same UMAP space, we next shifted our focus to further characterize osteoclast heterogeneity. Consistent with human and murine reports using scRNA-seq to characterize OCs, we identified 4 transcriptomically distinct OC populations[12,13,41]. The cycling OCs (c5/c8) in our canine OS dataset likely correspond to previously reported pre/progenitor OCs, while the mature OCs (c6) are consistent with previous reports (Fig. 6a, b). CD320[+] OCs (transcobalamin receptor expressing OCs, c9) had not been described in macrophage or osteoclast clusters from human and mouse tissues and may represent a canine specific cell type, or possibly a previously unresolved OC subtype (Fig. 6a, b). Due to the similarity of OCs and macrophages we completed hierarchical clustering to confirm the unsupervised clustering results (Fig. 6c). The secondary analysis was consistent with unsupervised clustering and further suggested Clusters 5, 6, 8, and 9 are distinct from the macrophage clusters.

To confirm the mature OC classification and provide a canine specific transcriptomic signature, we completed DGE analysis. When comparing mature OCs (c6) to macrophages (c0, c1, c2, c3) we identified canine mature OCs to be defined by *ATP6V1C1*, *CD84*, *HYAL1*, and *CAMTA2* expression, which subsequent GSEA analysis revealed an association with bone resorption and remodeling (Fig. 6d, Supplementary Data 3, Supplementary Fig. 13a). We next completed DGE analysis contrasting CD320[+] OCs (c9) with macrophages and mature OCs. (Supplementary Data 3, Supplementary Fig. 13b, c). By evaluating the intersection of the differentially expressed genes we determined CD320[+] OCs are defined by *HMGA1*, *TNIP3*, and *CD320* expression (Fig. 6e). The analysis also provided further evidence that c9 is an OC cluster based on *TNFRSF11A* (RANK) enrichment when contrasted with macrophage, but not when contrasted to mature OCs[42]. Lastly, we used pySCENIC's regulon specificity scoring to better define the transcription factors active in mature OCs and CD320[+] OCs. We identified *ZEB1* and *NFATC1*, known regulators of OC development, to be enriched in mature OCs, while *SNAI1* and *ETV3/6/7* were enriched in CD320[+] OCs suggesting a differentiating cell type (Fig. 6f, g)[43–45]. Together our analysis indicates, CD320[+] OCs are a distinct population from mature OCs that may represent an OC precursor.

### Transcript abundance of widely used immunohistochemistry macrophage markers exhibit distinct specificity to myeloid cells

In contrast to other tumor types, there have been multiple reports in humans and dogs suggesting that increased TAM infiltrates in OS are correlated with reduced metastasis rates and increased patient survival[8,9]. Despite these reports, other groups completing similar analysis have concluded that increased macrophage infiltrates have a negative impact on OS clinical outcomes[46]. Given the conflicting nature of previous reports we sought to employ our dataset to investigate which cell types express the transcript of the prototypical macrophage markers used for IHC analysis. To complete this analysis, we profiled TIMs, TAMs, DCs, and OCs for the expression of widely used canine (*MSR1* aka CD204 and *AIF1* aka Iba1) and human (*CD163* and *CD68*) macrophage markers (Fig. 7a). With the caveat that this analysis is limited to transcript abundance and does not evaluate protein expression, we found that *CD163* transcript expression was the most specific for macrophages. *CD68* expression was detected in TIMs, TAMs, and OCs, with remarkably high expression levels in mature OCs. The expression of *CD68* on mature OCs is consistent with human literature[47]. *AIF1* (Iba1) was the most non-specific marker with diffuse expression across all cell types, except for mature OCs. Lastly, *MSR1* (CD204) was determined to be largely specific to TAMs, but the expression also extended to CD320[+] OCs and CD4[+] monocytes. To investigate the translational relevance of this finding, we evaluated expression of the markers in human OS (Supplementary Fig. 14). We observed similar expression patterns, with marked variability in specificity of each marker, suggesting the variability is conserved across species.

Given the degree of heterogeneity within the myeloid compartment in the OS TME, we used a Wilcoxon Rank Sum test to identify features that define each cell type, then selected for features predicted to be expressed on the cell surface (Fig. 7b, Supplementary Fig. 15, Supplementary Data 5). Overall, the analysis suggested there is substantial overlap in expression of most features. Despite the overlap, we were able to identify candidate markers which included *ADAM28* for LA-TAM_C1QC[hi], *TNFSF13B* for IFN-TAMs, and *CD84* for mature OCs. Lastly, we calculated the relative percentages of each cell type to further facilitate cell identification (Fig. 7c, Supplementary Table 6). Together, the data presented here act as a foundation to further investigate the role of myeloid cells in OS biology.

### Cell-cell interaction analysis indicates TAMs are involved in immune regulatory pathways

Following cell identification through subclustering analysis of major cell types, we evaluated the cell-cell interaction networks using CellChat. Between the 41 cell types included in the analysis, we identified a total of 13,235 inferred interactions across 59 signaling networks. The number of interactions and the predicted interaction strength of incoming (express receptor) versus outgoing (express ligand) signals were used to infer the activity of cells within the TME (Fig. 8a, Supplementary Fig. 16a). The top three cell types predicted to have the strongest interactions were fibroblasts, mature OCs, and endothelial cells. We next categorized the significantly enriched networks as "immune specific", "immune related", and "non-

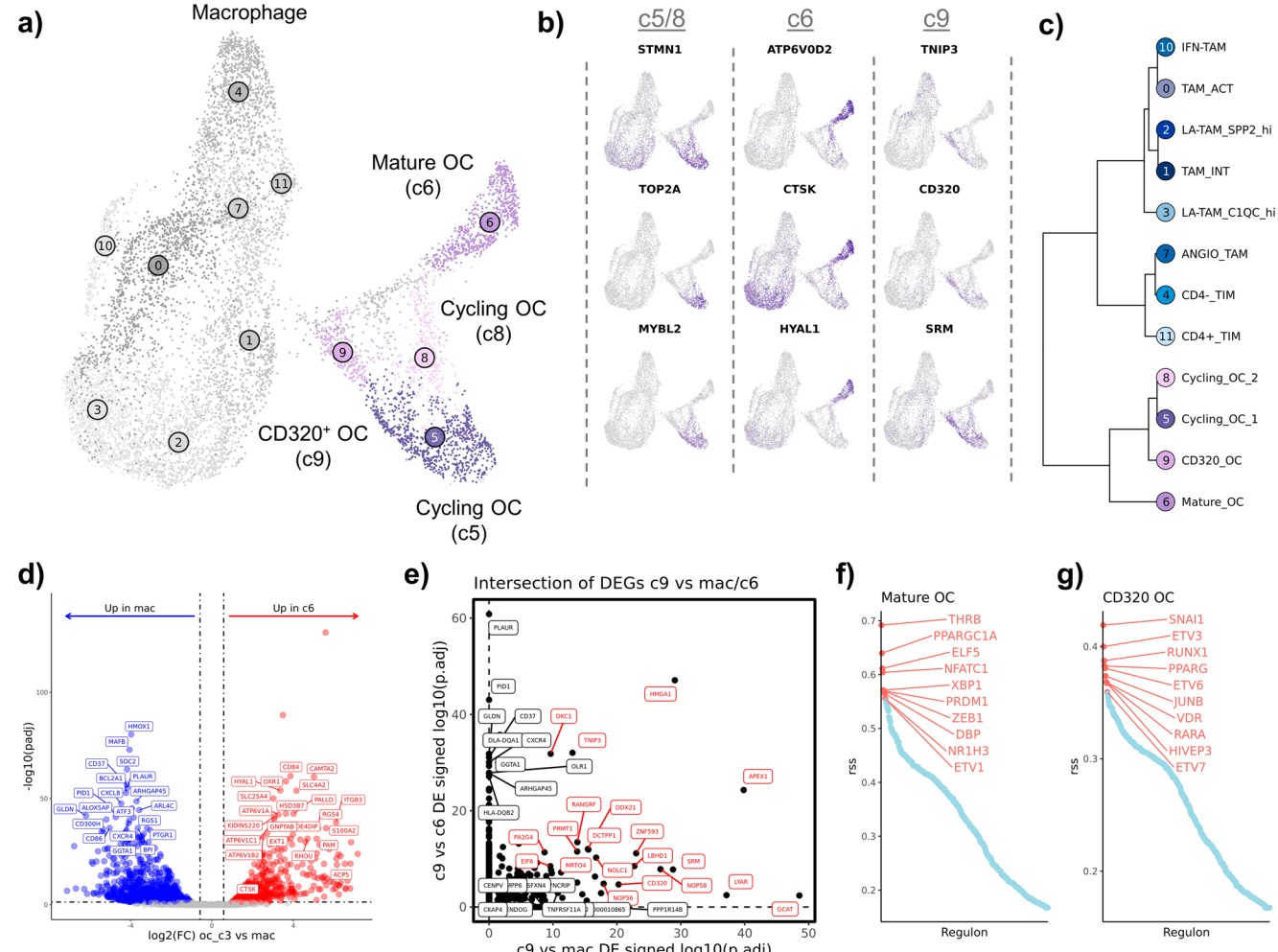

**Fig. 6 | Four transcriptionally distinct osteoclast subtypes are identified using unsupervised clustering. a** UMAP representation of macrophage and osteoclast (OC) populations (*n* = 10,512). The 8 macrophage subtypes defined in Fig. 5 are grayed out. **b** Feature plots of canonical features used to define cell types. (**c**) Dendrogram depicting results of hierarchical clustering of OC subtypes using cluster averaged log normalized expression. **d** Volcano plots depicting results of differential gene expression analysis when comparing mature OCs (c6) versus macrophages (c0, c1, c2, and c3). The top 20 features (weighted by adjusted *P* value) are labeled for each plot, *ACP5* and *CTSK* additionally labeled in **d**. **e** Scatter plot depicting the signed log10(adjusted *P* value) when contrasting CD320+ OC versus macrophage (c0, c1, c2, and c3) (x-axis) and CD320+ OCs versus mature OC (y-axis). The top 20 co-enriched features are labeled in red, while the top 7 features enriched in one species are labeled in black (**f,g**) Scatter plot depicting regulon specificity score (rss) for active transcription factors in mature OC (**f**) and CD320+ OC clusters (**g**), as determined using pySCENIC.

immune" to investigate if certain cell types were more active in a subset of networks (Fig. 8b). We found that malignant osteoblasts and stromal cells were largely predicted to be involved in "non-immune" interactions, while "immune specific" interactions were largely confined to TAMs and DCs with strong outgoing interactions.

By subsetting on immune cells and evaluating interactions of known immune regulatory pathways we identified mregDCs and IFN-TAMs to have the most interactions, while activated (*CD5L*+) macrophages and C1QC^hi LA-TAMs were predicted to have the strongest outgoing signals (Fig. 8c). It was further predicted that follicular helper and regulatory CD4 T cells make up the populations receiving most of the signals originating from myeloid cells. When evaluating the PD-L1 network, we identified mregDCs, TIMs, and IFN-TAMs to have the highest expression of PD-L1 and were predicted to interact with T_fh, T_regs, and exhausted CD8 T cells (Fig. 8d, Supplementary Fig. 16b). The CD80 and CD86 networks involved a larger portion of myeloid cells, with all CD4 T cells predicted to be influenced by the interactions (Fig. 8e, Supplementary Fig. 16c, d). Overall, activated TAMs, IFN-TAMs, and C1QC^hi LA-TAMs are predicted to be key contributors to shaping T cell mediated immunity.

## Comparison of human and canine scRNA-seq OS datasets reveal a high degree of similarity in cell type gene signatures between species

Lastly, we obtained 6 publicly available treatment-naïve human OS scRNA-seq samples to complete a cross-species analysis (GSE162454)[12]. The two datasets were integrated using a Seurat alignment workflow which is reported to overcome genomic annotation differences between species[48]. Hierarchical clustering of the integrated SCTransform normalized data revealed a high degree of similarity between species, with major clades containing similar cell types based on pre-integration annotations (Fig. 9a). Evaluation of similarities in cell type gene signatures using Jaccard similarity index produced similar results (Supplementary Fig. 17). All canine lymphocyte subtypes paired 1:1 with their human counterpart, as did endothelial cells and fibroblasts. Discrepancies between species included the placement of plasmacytoid dendritic cells (pDCs), which clustered into separate clades, and weak Jaccard similarity index values for pDCs and mast cells across species. Overall, macrophages clustered in the same clade, but due to differences in annotation levels, many cell types did not pair off into terminal clades.

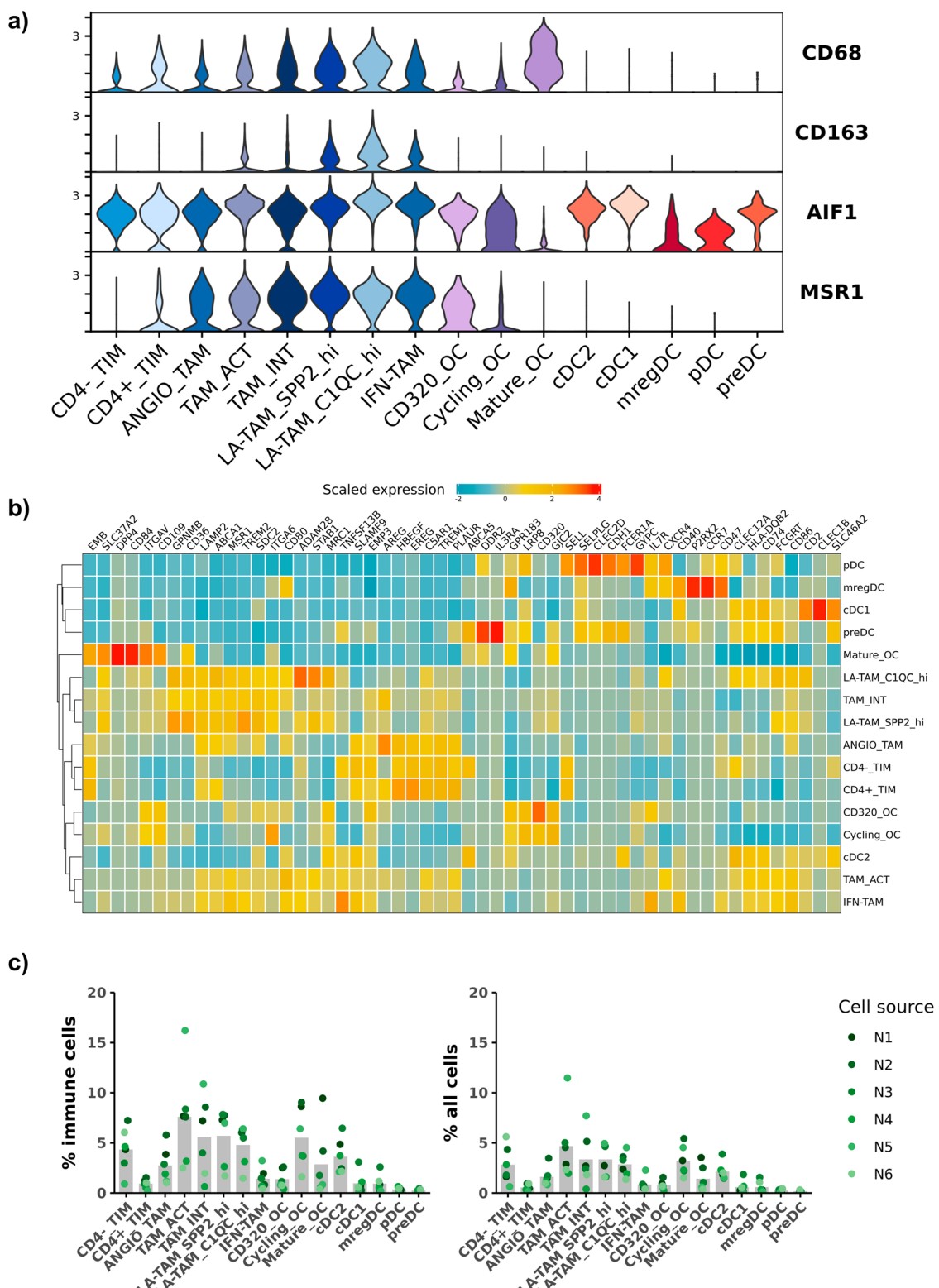

**Fig. 7 | Transcript abundance of widely used immunohistochemistry macrophage markers reveal variable specificity. a** Violin plots, grouped by canine cell type, depicting expression of four immunohistochemistry (IHC) macrophage markers widely used to evaluate macrophage infiltrates in canine and human osteosarcoma. **b** Scaled expression of the top 3–5 features for each cluster. The selected features were chosen based on the results of FindAllMarkers() and the inclusion of the feature in the surfaceome database listed as predicted surface expression. **c** Bar chart of the mean percent composition for each cell type as a percentage of total immune cells and percentage of all cells (osteoclasts were included as an immune cell in the calculation). Each dot corresponds to a biological replicate.

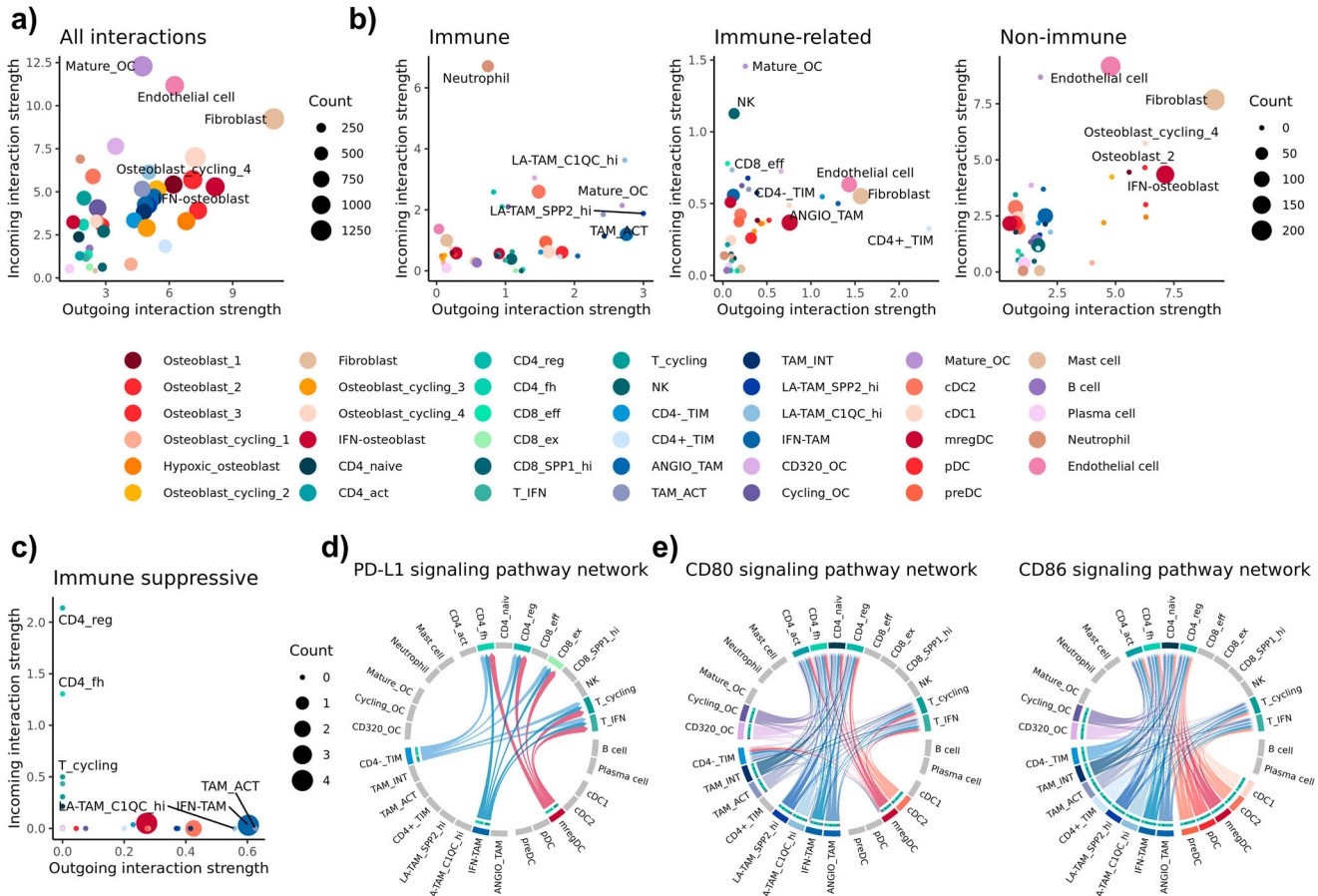

**Fig. 8 | Cell-cell interaction analysis reveals fibroblasts as a key communicating cell type and identifies TAMs as immune regulatory. a** Scatter plot depicting the strength of outgoing (x-axis) and incoming (y-axis) signals for all cell types and calculated using all enriched signaling networks. **b** Scatter plot depicting interaction strengths for three subdivided networks, "immune", "immune-related", and "non-immune". **c** Scatter plot depicting the interaction strengths for immune cells

calculated using three immune suppressive signaling networks (PD1/PDL1 & CD80-CD86/CTLA4). Circos plots of immune regulatory networks identified using CellChat with (**d**) PDL1 and (**e**) CD80/CD86 networks depicted. The arrow origin represents expression of a ligand, while terminal arrow indicates expression of a receptor. Cell types not involved in the network are grayed out.

To further compare transcriptional programs across species we used an analysis approach adopted from Scheyltjens et al.[49]. Briefly, the approach used DGE analysis between two cell populations in each species, then signing of the adjusted *P* value to determine if transcriptomic signatures were conserved. When contrasting fibroblasts and endothelial cells, we found substantial overlap in gene expression patterns with key endothelial cell markers (*PLVAP*, *CD34*, and *PECAM1*) enriched in both species (Fig. 9b, Supplementary Data 6). Top features conserved in fibroblasts included *VCAN*, *COL6A1*, and *LUM*, while key features such as *FAP* and *ACTA2* were also conserved. Interesting discrepancies included the expression of *HYAL2* and *NOTCH* as defining features in human endothelial cells, but nonsignificant in canine endothelial cells.

Completion of the same analysis on plasmacytoid DCs and cDC2s revealed *TCF4* to be enriched in pDCs and *BATF* expression enriched in cDC2s, which is consistent with human literature (Fig. 9c, Supplementary Data 7)[29]. An intriguing distinction between species included the high expression of *GZMB* and *PTGDS* (prostaglandin D2 synthase) in human pDCs, but not in canine pDCs. Lastly, we applied the same approach to compare mature OCs with TIMs (Fig. 9d, Supplementary Data 8). As expected, mature osteoclasts were defined by *CSTK*, *ACP5*, and *ATP6V0D2* expression, while monocytes in both species were defined by *CXCL8*, *OSM*, and *LYZ* expression. Notable differences included canine monocytes exhibiting high expression of *SLAMF9* and *PLBD1*, while human monocytes had high *S100A8* and *HCST* expression. In summary, we present a comprehensive comparison of human and canine OS cell types, which suggests a

high degree of consistency in cell type gene signatures across the two species, however we also present evidence of distinct transcriptional programs in pDCs, mast cells, and monocytes.

## Discussion

In the present study, we completed a comprehensive analysis of canine osteosarcoma (OS) using single-cell RNA sequencing which revealed the complex network of cells within the tumor microenvironment (TME). Through analysis of 6 treatment-naïve canine OS samples we were able to identify 30 distinct immune cell types, 9 unique malignant osteoblast populations, 1 cluster of fibroblasts, and 1 population of endothelial cells (Supplementary Data 1). We described the transcriptomic heterogeneity within malignant osteoblasts, identified cell types that have not been previously reported in dogs, and applied our data set to investigate the transcript abundance of widely used macrophage surface markers. Ultimately, the data presented here act as a molecular roadmap of the canine OS tumor microenvironment which provides canine specific cell type gene signatures that can be applied to guide immunological reagent development and further investigations of the canine OS TME.

Prior to this study, evidence of a conserved OS TME between humans and dogs was limited, with the most recent human-canine comparisons being presented in Mannheimer et al.[50]. To compare the human and canine OS TME more directly we obtained a publicly available scRNA-seq human OS dataset which enabled evaluation of the relative relatedness of cell types between species. The analysis revealed that lymphocytes exhibited the

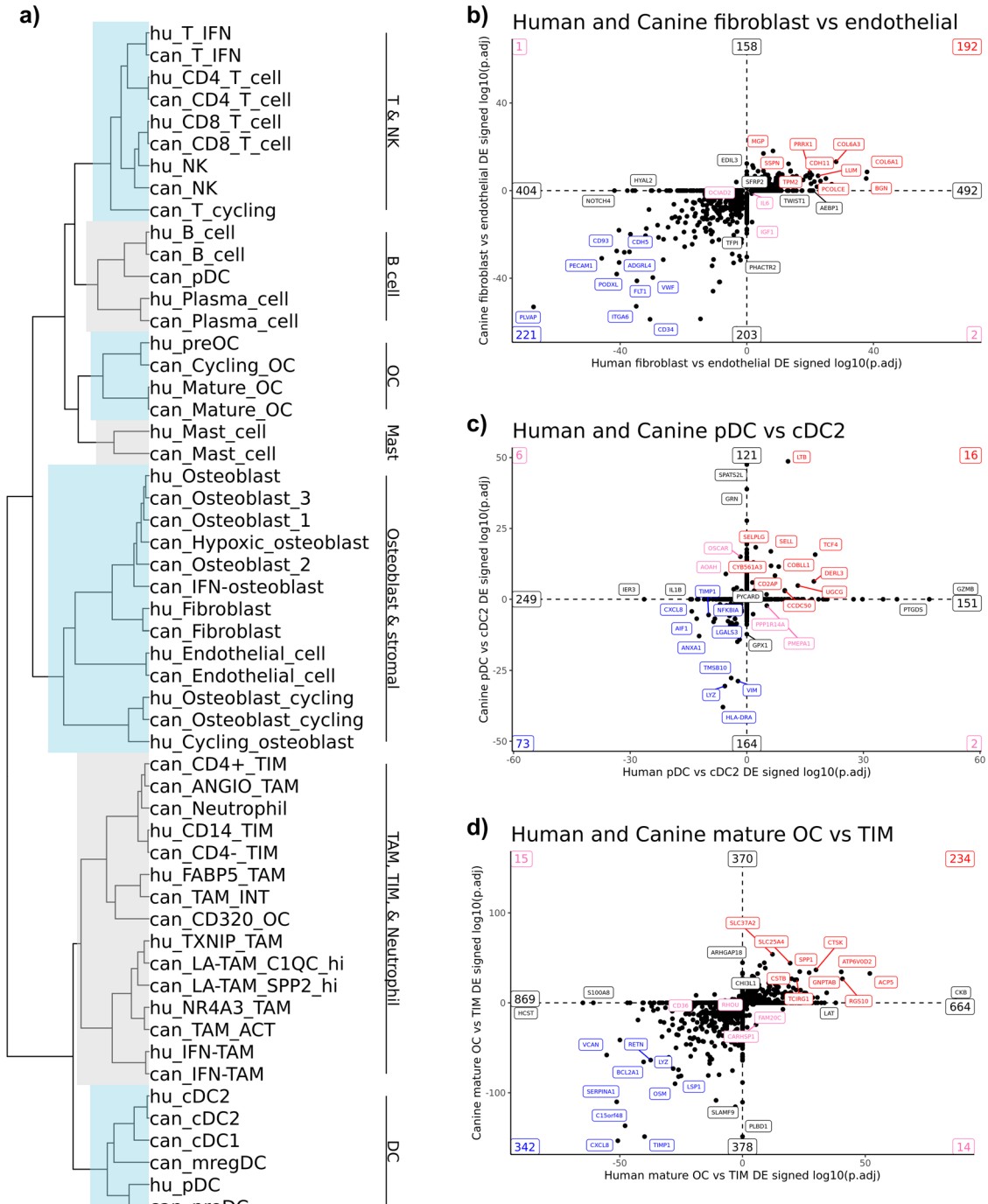

**Fig. 9 | Cell type gene signatures in osteosarcoma are conserved between human and canine. a** Hierarchical clustering of human ("hu_" prefix) and canine ("can_" prefix) cell types using SCTransform normalized data of a human-canine integrated dataset. Scatter plots comparing the signed log10(adjusted *P* value) of significantly upregulated or downregulated genes identified when comparing (**b**) human fibroblasts versus human endothelial cells and canine fibroblasts versus canine endothelial cells, (**c**) human pDC versus human cDC2s and canine pDC versus canine cDC2s, and (**d**) human mature OCs versus human CD14 monocytes and canine mature OCs versus canine TIMs (both CD4+ TIMs and CD4− TIMs).

In **b**–**d** conserved upregulated features are in the top right quadrant (top 10 in red labels) and conserved downregulated features are in the bottom left quadrant (top 10 in blue labels). Conflicting features – up in human but down in dog (bottom right quadrant) and down in human but up in dog (top left quadrant) – are labeled in pink (top 2 in each quadrant). Features up or down in one species, but not a differentially expressed gene in the other, fell on the axis and the top 2 on each axis direction are labeled in black. The numbers in the corners and at the ends of axis lines represent how many features fell in that region.

highest degree of conservation between species, while the gene signatures of major cell types also exhibited substantial overlap. The cell types with the most distinct transcriptional signatures included plasmacytoid dendritic cells and mast cells. These species differences could be the result of either distinct transcriptional profiles or, less interestingly, discordant annotations

between species. In both datasets plasmacytoid dendritic cells (pDCs) were defined by *FLT3*, *IGHM*, and *TCF4* which suggests consistent annotation in both species[51]. Therefore, the difference in gene signatures within pDCs may represent distinct transcriptional profiles. The mast cells identified in each species were defined by *GATA2* and *MS4A2* expression, but the population

had the weakest Jaccard similarity index suggesting poor conservation of gene signatures. *GATA2* (a transcription factor with implications in basophil and mast cell differentiation) and *MS4A2* (the IgE receptor found on basophils and mast cells) are not specific to mast cells, so it is possible that the cluster could instead represent basophils or possibly eosinophils[52–54]. Further functional and transcriptomic investigation of these cell populations is warranted.

Our analysis revealed the presence of many rare cell populations, including mregDCs, CD4$_{fh}$ T cells, and IFN-TAMs, opening avenues for further investigation of these cell populations. With the caveat that transcript expression may not correlate with protein expression, we used the surfaceome reference database to identify possible surface markers for further study of these cell types[27]. In addition to antibody-based assays, the transcriptomic signatures presented here provide a reference for the application of deconvolution algorithms (such as CIBERSORTx and TIMER) when evaluating bulk RNA sequencing data obtained from canine OS samples[55,56].

Mature regulatory dendritic cells represent a recently defined cell type which has been identified across several human tumor types, including OS[31,57]. The biological role of mregDCs is still being identified, but recent reports suggest a potential role in shaping T cell antitumor immune responses[24,25]. In our analysis, we were able to identify a *CCR7*[+]/*IL4I1*[+]/*FSCN1*[+] dendritic cell population which closely resembles the descriptions of human mregDCs. We found that canine mregDCs express high levels of immune suppressive and costimulatory molecules, which may play a role in modulation of adaptive immune responses through communication with follicular helper and regulatory T cells. This study provides evidence that mregDCs are present in canine OS, demonstrating a conserved role within osteosarcoma across species.

The heterogeneity within the myeloid compartment of OS is an area of intense interest, in particular the role of TAMs in regulating OS biological behavior and clinical outcomes is debated[58]. To provide further context for the discrepant findings reported in the literature regarding the role of TAMs, we evaluated the transcript abundances of key macrophage markers used in human and canine analysis. Although the analysis was completed at the transcript level, we observed notable differences in the specificity of each cell type marker within given myeloid populations. Inconsistencies between TAM cell types using immunohistochemistry (IHC) could explain why some groups identify negative prognostic correlates with TAM density, while other groups report that TAM density is correlated with positive outcomes[8,9,46]. Further validation of the variability in cell type markers could be completed using refined IHC panels coupled with spatial transcriptomics. Ultimately, a better understanding of the prognostic and functional roles of myeloid cells within the TME will aid in the development of effective myeloid cell targeted therapeutics.

While the single-cell RNA reference presented here provides key insights into canine OS, the dataset is not without limitations. First, although we sampled male and female dogs across a range of ages, our dataset still only consisted of 6 dogs and may not fully represent all cell populations found in canine OS. Secondly, the tumor sample obtained from one dog (dog 6) exhibited markedly more neutrophils relative to other samples which may suggest sample contamination with blood, bone marrow, or necrotic tissue. Lastly, cellular annotations largely relied on human gene signatures due to the lack of canine specific data available. This may have exaggerated similarities between species and may have resulted in the forcing of distinct canine-specific cell types into subtypes derived from human nomenclature. Thus, the discordant findings regarding mast cell gene signatures represents an important distinction that should be investigated further and considered when using the dog as a model for human disease.

The data presented here represent a valuable resource for comparative oncology research using the canine cancer model. A major goal of this project was to make the data accessible to the greater research community and multiple avenues are provided for researchers to explore and use the dataset (see data availability statement). Our comparisons between human and canine OS revealed the conserved nature of cell type gene signatures in

OS while also identifying potential differences. Overall, our analysis further supports the value of the dog as a model for human OS research and provides an important reference dataset to advance canine immuno-oncology research.

## Methods

### Study animals
Dogs included in the study were selected based on the presence of an appendicular primary tumor and the absence of previous therapeutic intervention. All dogs presented with radiographic evidence of OS and subsequent histopathological evaluation was completed to confirm the diagnosis. All study dogs underwent amputation of the affected limb and samples were collected for single-cell RNA sequencing processing within 30 min. The amputated limb was then submitted to the Colorado State University Veterinary Diagnostic Laboratory where representative samples of the tumor were processed for histopathologic confirmation of the clinical diagnosis. The osteosarcoma subcategorization presented in Table 1 was based on pathology reports for each sample and confirmed by a second veterinary pathologist. All studies were approved by the Colorado State University (CSU) Institutional Animal Care and Use Committee and the CSU Clinical Review Board. We have complied with all relevant ethical regulations for animal use and all dog owners provided informed consent prior to sample collection.

### Sample preparation
The amputated limb was dissected to the level of the tumor and a stainless-steel Michele trephine biopsy needle was used to obtain 3–5 biopsy cores from the tumor, targeting areas where there was an obvious mass effect and/or lysis of cortical bone. The biopsy cores were then washed with phosphate buffered saline (PBS), minced using a scalpel, and digested with collagenase type II (250 U/mL) in Hanks' Balanced Salt Solution (HBSS) for 45 min at 37 °C with agitation (Thermo Fisher Scientific Inc.). Samples were passed through a 70-µm cell strainer, washed with PBS, then centrifuged for 5 min at 400 rcf. Each of the separately collected biopsies were inspected to ensure the presence of viable cells (as determined using trypan blue exclusion; Thermo Fisher Scientific Inc.), then all samples with detectable live cells were pooled into 4-mL HBSS. To enrich for live cells and remove debris, the pooled cell suspension was layered onto 3-mL Ficoll Paque (Cytiva; Marlborough, MA), and centrifuged for 30 minutes at 400 rcf with acceleration at 9 and brake at 0. Following density centrifugation, the cell interface layer was collected, washed one time with PBS, and resuspended in 10-mL of Ammonium-Chloride-Potassium lysis buffer for 3–7 min at room temperature. To remove small debris and platelets, a final wash at 100 rcf for 15 min was completed. Cells were resuspended in 0.04% molecular grade BSA (Sigma-Aldrich; St. Louis, MO) in PBS, confirmed to have a viability greater than 90% (as determined using trypan blue exclusion), and transported to a Chromium iX instrument (10x Genomics; Pleasanton, CA) for cell capture. All samples were captured within 30 min of preparation.

### Library preparation and sequencing
Single cells were isolated and tagged with molecular barcodes using a Chromium iX instrument with a target of 5000 cells per sample. Two of the six dogs (dogs 1 and 2) had two samples processed each with a 5000-cell target, for a total target of 10,000 cells for dogs 1 and 2. Single cells were processed using a Chromium Next GEM Single Cell 3′ v3.1 Kit and a standard Illumina library dual index library construction kit (10x Genomics). Sample quality was analyzed using a LabChip (PerkinElmer; Waltham, MA) and submitted for sequencing on an Illumina NovaSeq 6000 sequencer (Novogene Corporation; Sacramento, CA) with a target of 100,000 150 bp paired-end reads per cell.

### Read mapping and quantification
A Cell Ranger analysis pipeline (version 6.1.2, 10× Genomics) was utilized to process raw FASTQ sequencing data, align reads to the canine genome, and generate a count matrix. The default settings were used when running

"cellranger count" and aligned to a CanFam3.1 reference (Ensembl release 104) prepared as previously described[21].

## Data filtering and integration

For each sample, the count matrix was imported into R using the Read10X() function then converted to a Seurat object using the CreateSeuratObject() function[18]. To estimate the number of dead/poor quality cells, the percentage of reads mapping to mitochondrial chromosomes per cell ("percent.MT") was calculated using PercentageFeatureSet() to count all reads mapped to features with the prefix "MT-". Each object was filtered to only retain cells which met the following requirements:  200 < nFeature_RNA < 5500, percent.mt < 12.5, and 100 < nCount_RNA < 75000. Next, DoubletFinder, was used to identify and remove putative cell doublets[59]. After completing quality control filtering on each sample, all samples were normalized using SCTransform (Pearson residuals of regularized negative binomial regression) then integrated into one object using Seurat's alignment workflow[48]. The alignment workflow consisted of (1) identification of variable features in each sample, (2) scaling data for variable features in each sample, (3) identification and filtering of conserved variable features between samples ("anchors") using canonical correlation analysis (2000 integration anchors used), and (4) pairwise integration of the samples. During the data scaling step, we used the "percent.MT" value as latent variable in a linear regression framework to minimize the impact of mitochondrial reads on dimension reduction and integration[60]. Following data integration, the dataset was inspected and three low quality clusters (defined by low UMI counts) were identified and removed from the dataset. The filtered dataset was then divided into a list of count matrices by sample and Seurat's alignment workflow was repeated, as the selection of variable features is potentially altered with removal of cells. Ideal clustering parameters (res = 0.8, dims = 45, n.neighbors = 40, min.dist = 0.35) were determined using the R package clustree[61]. Dimension reduction and visualization was completed, and the data were projected using 2-dimensional, non-linear uniform manifold approximation and projection (UMAP) plots.

## Subclustering analysis

For each major cell type, we subset the integrated dataset onto the population of interest to exclude all additional cells. The subset dataset was then divided into a list of count matrices by sample and Seurat's alignment workflow was repeated as described above. In the process of repeating the integration new variable features were identified which can enhance the ability to detect rare cell types through unsupervised clustering of the reintegrated dataset. The integration, dimension reduction, and clustering parameters were as follows; tumor/stroma: integration anchors = 3000, res = 0.5, dims = 40, n.neighbors = 50, min.dist = 0.5, T cell: integration anchors = 2500, res = 0.6, dims = 40, n.neighbors = 50, min.dist = 0.3, dendritic cells: integration anchors = 2000, res = 0.3, dims = 35, n.neighbors = 50, min.dist = 0.3, and macrophage/osteoclasts: integration anchors = 2500, res = 0.6, dims = 40, n.neighbors = 40, min.dist = 0.25. During subclustering analysis additional low-quality clusters (low UMI or heterotypic doublets) were filtered out. Specifically, 3 (tumor/stroma), 1 (T cell), 0 (dendritic cells), and 1 (macrophage/osteoclasts) cluster(s) were removed from each major subset.

## Cell classification

High level cell type annotations were established using unsupervised clustering results, gene set enrichment analysis, and manual annotation based on the literature for human cell type markers[62]. Briefly, features used to identify major cell populations included *CD3E/CD5/CD7* for T cells, *CTSK/ACP5/ATP6V0D2* for osteoclasts, *CD68/AIF1/MRC1* for macrophages, *S100A12/CD4/CXCL8* for neutrophils, *COL1A1/ALPL/FAP* for tumor/fibroblasts, *FLT3/CD1C/DNASE1L3* for dendritic cells, *MS4A1/JCHAIN* for B cells, *ESAM/PLVAP/CD34* for endothelial cells, *TOP2A/H1-5/MKI67* for cycling cells, and *FCER1A/GATA2/MS4A2* for

mast cells. In addition to the use of canonical markers, singleR and reference mapping to a canine leukocyte atlas was completed to provide support for immune cell classifications[21,63]. Further high-resolution cell identification was completed through subclustering analysis on cells within each major population (Tumor/fibroblast, macrophage/monocyte, osteoclast, dendritic cell, and T cell). Cell type gene signatures, as determined in this dataset using the FindAllMarkers() function (Wilcoxon Rank Sum test), can be found in Supplementary Data 1. In addition to the full gene signatures, we provide curated short cell type gene signatures in Supplementary Data 2 and Supplementary Table 1.

## Feature visualization

Feature expression was visualized using violin plots, feature plots, and dot plots. Selected features were chosen based on prior biological knowledge and classification of features as statistically significant using the FindAllMarkers() function. Y-axis scales for violin plots within a figure are on fixed scales. Feature plots show normalized expression for each feature on variable scales. For all feature plots, gray/light purple coloration indicates low expression and dark purple coloration indicates high expression. Dot plots use scaled expression data which depicts deviation from the average value for a gene across the cells being sampled.

## Differential gene expression analysis

Differential gene expression (DGE) analysis was completed using pseudo-bulk conversion followed by a DESeq2 pipeline[64]. Prior to running DESeq2, low abundance features, defined as features with less than 10 raw counts across all cells sampled, were filtered out. Features that had an adjusted $P$ value of less than 0.05 (as determined using a Benjamini and Hochberg correction method) and a log2(fold change) greater than 0.58 were considered to be statistically significant[65].

## Gene set enrichment analysis

When completing follow-up gene set enrichment analysis (GSEA) on the gene lists generated from DGE analysis, the significantly upregulated and downregulated features were processed separately. The upregulated and downregulated gene lists were used as input for evaluation using the clusterProfiler and msigdbr R packages to infer pathway activity[66,67]. Terms which reached an adjusted $P$ value of 0.05 or lower (Benjamini and Hochberg correction method) were discussed as significantly enriched.

In addition to using GSEA following DGE analysis we also used the R package singleseqgset to complete GSEA on cell type clusters. The analysis used a competitive gene set enrichment test which was based on a Correlation Adjusted MEan RAnk gene set test[68]. The log2(fold change) and mean expression for every feature within each cell type was calculated and used to complete GSEA. $P$ values were corrected for multiple comparisons using a false discovery rate (FDR) method and corrected $P$ values were filtered to only retain terms in which at least one cell type had a value less than 0.05. The enrichment values were scaled, and the top pathways (weighted by $P$ value) were plotted using a heatmap.

## Copy number variation analysis

Copy number variation (CNV) analysis was completed using CopyKAT on cells that contained more than 2000 unique molecular identifiers (UMIs)[14]. Briefly, the approach segmented the human genome into 220-kb variable genomic bins to establish a genome-wide copy number profile for each single cell at an approximate resolution of 5 Mb. Eash sample was run individually with a known normal cell population consisting of osteoclasts, neutrophils, macrophages, and T cells when inferring CNV status. Individual cell classifications were extracted from the CopyKat output files, transferred to the integrated dataset, and CNV status was visualized in the UMAP space. This approach was only used to infer if a cell was aneuploid or diploid. Individual chromosomal mutations were not evaluated due to incompatibilities of the software across species.

To supplement the CopyKAT analysis we also applied the inferCNV algorithm to the dataset which yielded similar results in terms of aneuploid

and diploid classifications[15]. Briefly, the inferCNV approach was run with an ordered gene file generated using the Ensembl CanFam3.1.104 gene transfer format (.gtf) file that was used to build the reference used for alignment of the single-cell data. An infercnv object was then created using the canine gene positions with endothelial cells and macrophages selected as the "normal" reference populations. InferCNV was then run using the default settings with the recommended "cutoff" argument set to 0.1.

## Regulon activity

The python implementation of Single-Cell Regulatory Network Inference and Clustering (pySCENIC) was used to infer activity of gene regulatory networks within cell types[69,70]. To complete this analysis the gene symbols were converted from canine to human using the convert_orthologs() function from the orthogene R package[71]. During conversion, genes that had duplicate mappings in either canine or human annotations were dropped from the matrix and excluded from downstream analysis. The count matrices with converted gene symbols were loaded into SCANPY and a standard pySCENIC workflow with default settings was followed[72]. The regulatory feather files used in the analysis were obtained from https://resources.aertslab.org/cistarget/, with file names being hg38__refseq-r80__10kb_up_and_down_tss.mc9nr.feather hg38__refseq-r80__500bp_up_and_100bp_down_tss.mc9nr.feather. After predicting regulon activity with pySCENIC, regulon specificity scores (rss) were calculated using AUCell and the rss values were used to infer regulon activity in the cell types analyzed[73].

## Cell-cell interaction inference analysis

The R package, CellChat, was used to make inferences about cell-cell interactions within the tumor microenvironment[35]. Using a list of known human receptor-ligand pairs provided through CellChat, we calculated the interaction scores (strength and weight) which represent the probability of two cells interacting. Analysis was initially completed on the final fully annotated dataset with 41 cell types and secondarily completed on a subset of only the immune cells (exclusion of the major cell types: osteoblast, cycling osteoblast, endothelial, and fibroblast). Prior to analysis the raw count matrices were extracted and the gene symbols were converted from canine to human as described in the "Regulon activity" section. Broadly the analysis evaluated the interactivity between ligand expressing cells (senders/outgoing signals) and receptor expressing cells (receivers/incoming signals). Inferences regarding potential interactivity were then made based on the law of mass action using the average expression values of receptors and ligands within cell types. Statistical enrichment of interaction networks was determined using permutation testing and an adjusted $P$ value < 0.05 was used to determine significance. Following identification of enriched cell-cell interaction networks, we further classified each enriched pathway as "immune specific", "immune related", and "non-immune" based on reported expression patterns of receptors and ligands (Supplementary Table 2). Analysis of the immune cell subset involved further classifying a subset of interaction networks (PD-L1, CD80, and CD86) as immune regulatory.

## Human OS homology analysis

Six treatment-naive human OS samples were obtained from the NCBI GEO database accession GSE162454[12]. The count matrices reported from the previous study were loaded in as Seurat objects and were filtered using the same parameters as used to preprocess the 6 canine OS tumor samples. The human dataset was annotated using low resolution unsupervised clustering while referencing the primary article in an attempt to recreate the original annotations. Prior to integrating data across species, the raw count matrices from each dataset were extracted and canine gene symbols were converted from canine to human as described in the "Regulon activity" section. Following conversion of gene symbols, the 12 OS samples (6 human and 6 canine) were integrated into one object using the same Seurat alignment workflow described above, with the exception that 3000 variable features were selected as anchors. SCTransform normalized counts were then used

to complete hierarchical clustering using the hclust() function with method set to "complete". Subsequent cell type gene signatures were established using FindAllMarkers() and DGE analysis contrasting cell types within each species was completed within the respective dataset. The resulting cell type gene signatures were used to calculate a Jaccard similarity index, whereas the adjusted $P$ values were assigned a sign $(+/-)$ based on the log2(fold change) then the signed $P$ values were used to generate scatter plots.

## Statistics and reproducibility

Raw data from a total of 8 canine scRNA-seq osteosarcoma samples were generated in this study. Two of the 8 samples were technical replicates, which were considered as one sample when completing computational analysis to retain a total of 6 biological replicates. Biological replicates were used for pseudobulk differential gene expression analysis, while cellular replicates were used for all other analysis completed in this study. Detailed descriptions of the statistical analyses and significance thresholds used in this study are provided in the respective methods section.

## Reporting summary

Further information on research design is available in the Nature Portfolio Reporting Summary linked to this article.

## Data availability

Raw sequencing data are available on the NCBI Gene Expression Omnibus database under the accession number GSE252470. The annotated dataset is available for browsing at the UCSC Cell Browser (https://cells.ucsc.edu/?ds=canine-os-atlas)[74], and the processed data (Seurat v4.3.0 RDS objects) are available on Zenodo (https://doi.org/10.5281/zenodo.10666968)[75]. Results of all differential gene expression analysis and cell type gene signatures are provided in Supplementary Data files.

## Code availability

A project specific GitHub page containing all analysis code and software versions used to analyze the data presented in this manuscript is available at https://github.com/dyammons/canine_osteosarcoma_atlas[75]. Any additional data requests can be made by contacting a corresponding author.

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

## Acknowledgements
This project was supported by grants from the National Institutes of Health (NIH): U01 CA224182 (to S.D.) and T32 OD012201 (to D.T.A.), the Boettcher Foundation Webb-Waring Biomedical Research Award (to D.P.R.), and the Shipley Family Foundation (to S.D.).The authors would like to acknowledge Lynelle Lopez, Kara Hall, and Allister Aradi for their assistance with management of clinical cases and sample collection. This work utilized the Alpine high performance computing resource at the University of Colorado Boulder. Alpine is jointly funded by the University of Colorado Boulder, the University of Colorado Anschutz, and Colorado State University. Data storage was supported by the University of Colorado Boulder 'PetaLibrary.

## Author contributions
Conception and design: D.T.A., S.D. Experimentation and data acquisition: D.T.A., J.K., D.P.R. Data analysis and manuscript drafting: D.T.A., L.S.H., K.E.C., D.P.R., S.D. Final approval of completed manuscript: All authors.

## Competing interests
The authors declare no competing interests.
