## [Peer Review File · Communications Biology]

Reviewers' comments:

Reviewer #1 (Remarks to the Author):

The authors present a comprehensive study of single cell transcriptomes of the tumor micro environment of 6 treatment naive dogs (35,310 cells) with osteosarcoma. Relatively little single cell data has been reported for dogs, especially for tumor samples, and this is the first, to my knowledge for osteo. Study samples, sequencing data, and analysis seem to have been performed carefully and robustly with relatively standard approaches. A number of immune cell types, tumor cell populations and other cell types are successfully identified. These are further systematically characterized by a series of reclustering, differential gene expression, gene set enrichment, and cell-cell interaction analyses. Finally, cell typing results are systematically compared to human data to confirm they are largely concordant with some interesting differences. The paper is largely descriptive of the different cell populations identified, how they compare to each other and against published expectations. Nevertheless, this provides a useful preliminary reference of the dog tumor microenvironment for OS and will be a valuable dataset to the community for re-analysis. Other efforts are made to provide valuable reference information such as potential surface markers for defining Tregs, CD4fh and Tifn-sig phenotypes. The authors should be commended for this. I have only a few major and minor comments.

Major comments.

1. The use of CNV alterations (aneuploidy) to define malignant/tumor subclusters is a reasonable approach. However, it is unclear from the results how well this is working, especially on a sample by sample basis. While figure 1E does show convincing overlap between aneuploid cells and two well-defined clusters I would be curious to see this for each sample. Are there some samples that were "quieter" with respect to CNV for which tumor cells could not be easily identified. Were these tumor samples exome sequenced? If so, even a few somatic mutations detected at scRNA level could help confirm each samples tumor cluster.
2. Currently data depositions to GEO and UCSC browser are listed as TBD. All primary raw sequence data should be deposited to SRA at minimum. Secondary data deposited to GEO/UCSC would be a bonus. There are methods to deposit data in SRA with an embargo data and reviewer access. This deposition should be completed and verified before publication as this is one of the primary values of the work to the community.
3. In the introduction and discussion it is stated that this paper will help to overcome the reagent and technical limitations associated with using the dog as a model. It seems everything with was done with standard/commercially available reagents. Please explain/support this claim better or remove.

Minor concerns/questions.

1. Six samples and 35,000 cells is not really sufficient to be considered an "atlas". I would suggest rephrasing as a preliminary survey or similar. Future work from this group, or a consortium of those working on dog scRNA might yet produce an atlas.
2. Lines 138-139. It is mentioned that "three low quality clusters were identified and removed ... ", please elaborate on how this was done.
3. Line 166. A log fold change (FC) greater than 0.58 was considered statistically significant (along

with p-value cutoff). How was this FC cutoff selected? This should be justified.

4. Line 207-209. Check grammar of sentence.

5. Lines 209-210. It is described that "Only features with homologues across both species were used for integration". Do you mean orthologs? How were these defined? Orthologs according to what analysis/resource? How were one-to-many or many-to-one orthologs handled?

6. Lines 255-256. When the tumor/fibroblasts were further sub-clustered into cycling tumor, non-cycling tumor and fibroblast, how was cycling status determined? I missed this in methods. How do the individual tumor clusters align with different dog identities? Do some of these clusters correspond to individual tumors?

7. Lines 457-458. Check grammar of sentence.

8. Lines 506-507. Check grammar of sentence. "similarities" should not be plural?

Reviewer #2 (Remarks to the Author):

The authors report on the cellular and molecular heterogeneity of treatment-naive primary osteosarcoma (OSA) in the canine using scRNAseq. The study was more descriptive than hypothesis driven as their focus was to generate an scRNAseq atlas of the OSA tumor microenvironment (TME) in six untreated canine OSAs. Their stated future purpose was to inform identify OSA TME features and facilitate future study of canine OSA. To this end the authors collected tissue samples within 30 min from dogs undergoing amputation for OSA.

Critique: In reviewing this manuscript, the reviewer has concentrated on the quality and reproducibility of the methods and by inference the results. Key QC criteria should include the quality of sample collection and the RNA quality. The authors should describe how the samples collected and from what site in the OSA, for examples did a pathologist described the degree of necrosis or degree abnormal bone. While the samples were subcategorized in Table 1 was this based on the sample submitted for sequencing or on the H&E section taken by pathology after the fact? OSA is highly heterogeneous and sections can vary in cell type.

The samples (3-5 per patient) were processed following standard sample procedures for sc-RNAseq before processing using the 10X Chromium platform. Library preparation and sequencing followed standard procedures.

Critique: QC data (web summary) such as estimated number of cells, median reads per cell, reads mapped confidently to the transcriptome were not provided in supplementary data. In addition, the RNA integrity number (RIN) was not reported for the samples. in the data filtering section, the sentence starting " During this step, we regressed out the percent mitochondrial" provide more information about the term regressed.

In the Cell Classification section cell classification was based on a single was based on a single marker e.g. CD68 for macrophages etc. Critique: While some markers for canine immune cells are lacking it seems presumptuous to use a single marker to identify cells e.g. while FLT3 is important DC cells is also present on hemopoietic cells. In the CNV section the authors refer to high quality cells containing 2000 UMIs, how were these cells identified as highly quality without providing QC data. In the human OS homology section, the authors need to explain the sentence "Following annotation of each species, the 12 (6 human and 6 canine) OS samples were integrated into one object to using a SCTransform

workflow with 3000 variable features as anchors (sic)" with reference to integration of data using SCTransform which is used for normalization and variance stabilization of data in the Seurat package.

If the QC data is up to standard then the result drawn by the authors are important. The study is a first in kind in canine OSA. Nevertheless, to verify conclusions drawn from study will require spatial analysis. This because of heterogeneity of the OSA samples and while the study is scRNAseq it comes from a combination of cells i.e. a bulk analysis of the tumor. The authors do acknowledge limitations.

Reviewer #3 (Remarks to the Author):

Single-cell RNA sequencing reveals the cellular and molecular heterogeneity of treatment-naïve primary osteosarcoma in dogs.

Ammons et al. here have presented a comprehensive scRNA-seq atlas of canine osteosarcoma. Study represents an important resource for comparative genomics of canine cancers, especially from a point of charting an otherwise elusive canine tumor immune-microenvironment (TMEN). Authors do acknowledge a limited sample size of canine OS samples but that said, all of scRNA-seq analyses are rigorous and well-documented. This study highlights immune cell heterogeneity in terms of cell subtypes and their potential marker genes which could be of value for future immunotherapy trials involving canine osteosarcoma cases. Authors go beyond conventional cell type classification to infer in silico gene regulatory and surfaceome marker activity, the latter being of a potential value - by targeting highly conserved surface tumor antigens - in designing experimental CAR-T therapies in an otherwise unchanged treatment modalities for osteosarcoma patients. Study should be treated as a useful database or resource for cross-species comparison of TMEN and as authors stated, can be of value for future canine cancer studies to deconvolve bulk RNA-seq data using CIBERSORT (which otherwise is not feasible without canine-specific scRNAseq data). I have two comments which - if feasible within timeline and resources - can aid in improving rigor of cell type classification and provide experimental validation of a few key predicted cell types.

1. Since core cell type classification relies on UMAP, worth considering an orthogonal approach like CONCORDEX to validate cell type assignments from scRNA-seq data.

Jackson K, Sina Boeshaghi A, Gálvez-Merchán Á, Moses L, Chari T, Pachter L. Quantitative assessment of single-cell RNA-seq clustering with CONCORDEX. bioRxiv 2023:2023.06.28.546949. <https://doi.org/10.1101/2023.06.28.546949> and related thread: <https://twitter.com/lpachter/status/1692310091901616312>

2. A complementary approach to validate a few if not all cell types would be to carry out IHC based on predicted cell markers for scRNA-seq defined cell types, and validate if expression patterns are for such markers is concordant with scRNA-seq dot plots and restricted to a given cell types.

We appreciate the thought and effort that was put into the review of our manuscript, and we believe the comments have improved the overall impact and clarity of information conveyed by the manuscript. In working through the revisions, we made additional changes related to suggestions mentioned by the Reviewers that we wanted to bring to your attention. These changes included:

1. Addition of methods for CellChat which we unintentionally omitted [line 245-263].
2. Addition of “Independent reclustering” method section to further explain how each subtype was handled [line 153-166].
3. A switch from using of R implementation of SCENIC to the python implementation (pySCENIC) which is more scalable and uses a more recent dataset of regulatory networks (this led to subtle changes to the results) [line 230-244;425-426;506-509].
4. Updating of all analysis approaches that used human gene signatures. This included the human-canine similarity analysis (Figure 9), CellChat (Figure 8), and pySCENIC analysis (Figure 6 f/g, Supplemental figure 12b). Initially we did not properly convert canine gene symbols to human, so the revised approach should be more accurate, although no conclusions were altered with this change [described in line 232-236].

All lines with changes within the manuscript are marked with a vertical grey bar in the left margin. Below is a point-by-point response to the Reviewer’s remarks.

Reviewer #1 (summary):

The authors present a comprehensive study of single cell transcriptomes of the tumor micro environment of 6 treatment naive dogs (35,310 cells) with osteosarcoma. Relatively little single cell data has been reported for dogs, especially for tumor samples, and this is the first, to my knowledge for osteo. Study samples, sequencing data, and analysis seem to have been performed carefully and robustly with relatively standard approaches. A number of immune cell types, tumor cell populations and other cell types are successfully identified. These are further systematically characterized by a series of reclustering, differential gene expression, gene set enrichment, and cell-cell interaction analyses. Finally, cell typing results are systematically compared to human data to confirm they are largely concordant with some interesting differences. The paper is largely descriptive of the different cell populations identified, how they compare to each other and against published expectations. Nevertheless, this provides a useful preliminary reference of the dog tumor microenvironment for OS and will be a valuable dataset to the community for re-analysis. Other efforts are made to provide valuable reference information such as potential surface markers for defining Tregs, CD4fh and Tifn-sig phenotypes. The authors should be commended for this. I have only a few major and minor comments.

Major comment 1:

The use of CNV alterations (aneuploidy) to define malignant/tumor subclusters is a reasonable approach. However, it is unclear from the results how well this is working, especially on a sample-by-sample basis. While figure 1E does show convincing overlap between aneuploid cells and two well-defined clusters I would be curious to see this for

each sample. Are there some samples that were "quieter" with respect to CNV for which tumor cells could not be easily identified. Were these tumor samples exome sequenced? If so, even a few somatic mutations detected at scRNA level could help confirm each samples tumor cluster.

Response:

- Our limited interpretation of the CNV analysis stems from the fact that the most widely used single-cell CNV inference methods (CopyKAT and inferCNV) have only been validated for use in human and mouse. The developers do not support use in other species, so we did not want to overemphasize any findings and opted to only use the approach to provide evidence of tumor (ANEUPLOID) or non-tumor (DIPLOID) cells.
- Given the unsupported nature of CNV inference tools used in our report, we ran both CopyKAT and inferCNV analyses. Both approaches yielded consistent results in identifying the suspected tumor populations as ANEUPLOID. We chose to present the CopyKAT results because their algorithm outputs a binary labeling scheme (ANEUPLOID vs DIPLOID) which better illustrated the differences we wanted to convey [line 220-229].
- It is important to note that inferCNV considers the genomic locations features within the canine genome whereas CopyKAT uses either human or murine gene positions (we applied human). As such, we believe the results should be interpreted cautiously in dogs. If any approach would allow for a more refined analysis of CNV burden and approximate locations within the canine genome it would be inferCNV. We now include the results of inferCNV for each sample as supplemental figures (Supplemental figure 2-9).
- In the results added to the supplemental figures, we notably observed nearly identical results for the two dogs for which we completed technical replicates (ran two samples through the Chromium iX), which suggests the results are consistent between runs.
- Presented in the figure below is further evaluation of CNV burden on a by-sample basis in the UMAP space (this figure is not included in the manuscript).
- Using the inferCNV output, we estimated the CNV burden on cell-by-cell basis by summing the absolute value of $1 - (\text{inferCNV CNV estimate})$ for each feature within each individual cell. From there, we split the CNV burdens into quartiles calculated on a by-sample basis.
- The inferCNV analysis is consistent with CopyKAT analysis in identifying the tumor cells as having the highest CNV burden, while the small cluster of fibroblasts and non-tumor cells were predicted to have lower CNV burdens.
- Looking at the distribution of CNV burden across the tumor cells, there appears to be evidence that clustering and CNV status are related. When we projected the CNV quartiles onto the tumor/fibroblast subset we found osteoblast_3 (Cluster 2 of primary Figure 2) to have a lower CNV burden than other tumor cell clusters (green population in lower figure from sample N1). It is also possible that the osteoblast_3 cluster may represent normal osteoblasts, rather than malignant cells.
- Although this is a potentially interesting finding regarding osteoblast_3, without whole exome sequencing we have no method to validate the CNV predictions and believe the results should be left at classifications of cells as ANEUPLOID or DIPLOID.

Major comment 2:

Currently data depositions to GEO and UCSC browser are listed as TBD. All primary raw sequence data should be deposited to SRA at minimum. Secondary data deposited to GEO/UCSC would be a bonus. There are methods to deposit data in SRA with an embargo data and reviewer access. This deposition should be completed and verified before publication as this is one of the primary values of the work to the community.

Response:

- We apologize for not having the data properly deposited prior to review. We have uploaded the raw data (BioProject PRJNA1060701) and cell by gene count to matrices (NCBI GEO [GSE252470](https://www.ncbi.nlm.nih.gov/geo/query/acc.cgi?acc=GSE252470)).
- The data can be reviewed using the reviewer token: yngpacmqjwlnev through the GEO portal.
- We will also upload the final processed/annotated Seurat objects (.rds files) associated with the project to NCBI GEO project and/or to Zenodo for easy download. In addition to depositing the processed Seurat objects we will provide detailed instructions for downloading and using the data which will be available on our project associated GitHub page (https://github.com/dyammons/canine_osteosarcoma_atlas/tree/main/input)
- Data will be made publicly available for exploring through an interactive browser hosted by UCSC or CELLxGENE (this will be available at time of manuscript acceptance).

Major comment 3.

In the introduction and discussion it is stated that this paper will help to overcome the reagent and technical limitations associated with using the dog as a model. It seems everything with was done with standard/commercially available reagents. Please explain/support this claim better or remove.

Response:

- These statements were intended to indicate that there are species-specific reagent limitations (eg, a lack of cross reactive or canine specific antibodies; or the time and reagent challenges associated with developing qRT-PCR primer sets one by one) that hamper immune-oncology research in dogs. The datasets and the interactive cell browser generated here will provide a roadmap for new targets for canine antibodies or RT-PCR based approaches for defining new cell types. Additionally, the reference data generated here provides the necessary data to generate a canine specific reference for the deconvolution of bulk RNA-seq OS datasets as an alternative approach to evaluating cell abundances using conventional flow cytometry or IHC.
- The text has been updated to further clarify the point we are attempting to make [line 63-65; 613-615].

Minor Concern 1:

Six samples and 35,000 cells is not really sufficient to be considered an "atlas". I would suggest rephrasing as a preliminary survey or similar. Future work from this group, or a consortium of those working on dog scRNA might yet produce an atlas.

Response:

- The mention of atlas has been removed. We opted to use "reference" in most instances.

Minor Concern 2:

Lines 138-139. It is mentioned that "three low quality clusters were identified and removed ...", please elaborate on how this was done.

Response:

- The methods have been updated to increase transparency regarding our handling of low-quality cell clusters [line 145-147; 153-166].
- Below are additional details and justification for our handling decisions.
- The three low quality clusters were removed informatically following completion of an initial round of unsupervised clustering, although we initially took measures to remove low quality cells from the dataset earlier in our pipeline...
 - "Each object was filtered to only retain cells which met the following requirements: $200 < nFeature_RNA < 5500$, $percent.mt < 12.5$, and $100 < nCount_RNA < 75000$. Next, DoubletFinder, was used to identify and remove putative cell doublets"
- ...these thresholds were set leniently, as generally recommend (1, <https://www.10xgenomics.com/resources/analysis-guides/common-considerations-for-quality-control-filters-for-single-cell-rna-seq-data>), with the idea that we will have to be aware of additional low-quality clusters. These clusters were removed on the basis of having low $nCount_RNA/nFeature_RNA$ suggesting they were empty droplets (ambient RNA partitioned in a 10x GEM). Rather than increase the low end of the $nFeature_RNA$ filter we decided to remove the contaminating low quality clusters. The reason for this choice was because raising the $nFeature_RNA$ threshold inadvertently removed neutrophils due to their naturally smaller transcriptomes.
- If interested, see lines 42-91 of the Figure1.R script for the code used to remove the cells. (https://github.com/dyammons/canine_osteosarcoma_atlas/blob/main/analysisCode/figure1.R)
- Additional data to support our claim that the clusters had low $nCount_RNA$ and $nFeature_RNA$ is presented below. The labeled UMAP presented below (top of page) depicts the pre-filtered data. The three clusters removed were c1, c11, and c28. We completed a preliminary analysis with these cells included, but they could not be assigned a biologically relevant cell type (a subjective assignment), and we observed low $nCount_RNA$ and $nFeature_RNA$ in the clusters (lower figure). In turn, we opted to remove the clusters as they are likely empty droplets.

nCount_RNA

nFeature_RNA

percent.mt

Minor Concern 3:

| *Line 166. A log fold change (FC) greater than 0.58 was considered statistically significant (along with p-value cutoff). How was this FC cutoff selected? This should be justified.*

Response:

- The threshold of a log fold change greater than 0.58 corresponds to a 1.5-fold change in magnitude between conditions. The P value threshold selected for use in this manuscript was 0.05 after correction for multiple comparisons.
- Commonly used thresholds for log₂(FC) include 0, 0.58, and 1; while commonly used P value thresholds include 0.01, 0.05, and 0.1 (2,3, https://hbctraining.github.io/DGE_workshop/lessons/05_DGE_DESeq2_analysis2.html).
- We selected moderately stringent thresholds prior to completing statistical analysis as we believed these thresholds would be appropriate for the analysis we presented in this manuscript.
- Complete gene lists resulting from all differential gene expression analysis performed in this manuscript with corresponding P values (adjusted and unadjusted) and log₂(FC) values are provided in supplemental data. As an aside, all significant DEGs were well above our chosen log₂(FC) threshold, so using a log₂(FC) > 1 instead of log₂(FC) > 0.58 would have no impact on interpretation of the data.
- We added a citation to Squair et al. where the authors investigated the prevalence of false positives in single-cell analysis and discuss best practices [line 197] (4).

Minor Concern 4:

| *Line 207-209. Check grammar of sentence.*

Response:

- The sentence was revised for clarity in response to this comment and one from Reviewer 2.

Minor Concern 5:

| *Lines 209-210. It is described that "Only features with homologues across both species were used for integration". Do you mean orthologs? How were these defined? Orthologs according to what analysis/resource? How were one-to-many or many-to-one orthologs handled?*

Response:

- We intended to refer to orthologs. The sentence in line 209 was updated to "Only features with 1:1 orthologs across..."
- In the initial submission we only included features that had identical gene symbols across species and omitted any gene symbols found in only one species.

- Given that this approach occasionally led to the omission of key orthologs from downstream analysis (i.e. “DLA-DRA” to “HLA-DRA”) we modified the approach in the resubmission.
- In the modified approach we used the `convert_orthologs()` function from the orthogene R package to identify 1:1 orthologs then converted canine gene symbols to human gene symbols. In the process of conversion, we dropped genes that had duplicate mappings in either canine or human annotations [line 232-236]
- This change led to the updating of Figure 8 and Figure 9. No main conclusions were altered as a result of this change, but the clustering of mast cells in the hierarchical clustering of human and canine cell types was altered such that they now pair off. We believe that this modification is a more appropriate approach to handle feature orthologs.
- We also added a supplemental figure depicting the Jaccard similarity index values calculated based on overlap in cell type gene signatures (Supplemental figure 17). This analysis (not included in the initial submission) continued to provide evidence of a weak overlap between mast cells, so we felt it was now important to include in the revised manuscript [method briefly described in line 278-279].

Minor Concern 6:

Lines 255-256. When the tumor/fibroblasts were further sub-clustered into cycling tumor, non-cycling tumor and fibroblast, how was cycling status determined? I missed this in methods. How do the individual tumor clusters align with different dog identities? Do some of these clusters correspond to individual tumors?

Response:

- Cycling status was largely determined based on the overexpression of cell cycle associated features (TOP2A, H1-5, MKI67) in conjunction with gene signatures similar to those observed in other tumor cell clusters present in our dataset.
- In addition to using select canonical markers to identify cycling cells, we also used the `CellCycleScoring()` function from Seurat which uses G2/M and S gene signatures defined in Tirosh et al. 2015 to predict which cells are likely actively cycling (5). The results are depicted below (next page), where cells labeled G2M or S would be expected to be cycling while G1 would be predicted to be resting.
- We did not observe any overt distribution skews in the tumor cell subset analysis. We added a supplemental figure to show distribution by sample (Supplemental figure 10a).

Minor concern 7:

| Lines 457-458. Check grammar of sentence.

Response:

- This has been corrected.

Minor concern 8:

| Lines 506-507. Check grammar of sentence. "similarities" should not be plural?

Response:

- "similarity" is more appropriate and has been updated.

Reviewer #2 (summary):

The authors report on the cellular and molecular heterogeneity of treatment-naive primary osteosarcoma (OSA) in the canine using scRNAseq. The study was more descriptive than hypothesis driven as their focus was to generate an scRNAseq atlas of the OSA tumor microenvironment (TME) in six untreated canine OSAs. Their stated future purpose was to inform identify OSA TME features and facilitate future study of canine OSA. To this end the authors collected tissue samples within 30 min from dogs undergoing amputation for OSA.

Critique 1:

In reviewing this manuscript, the reviewer has concentrated on the quality and reproducibility of the methods and by inference the results. Key QC criteria should include the quality of sample collection and the RNA quality.

Response:

- To our knowledge, traditional RNA integrity numbers (RINs) are not able to be evaluated from single-cell samples processed using the 10x Chromium iX platform due to the immediate conversion to cDNA in the GEMs.
- Library QC was evaluated using a LabChip prior to sequencing. All samples had a broad, rounded peak centered around 450 bp (see representative plot below). This distribution is expected based on the standard 10x protocol (see page 46 of user manual https://assets.ctfassets.net/an68im79xiti/1eX2FPdpeCgnCJtw4fj9Hx/7cb84edaa9eca04b607f9193162994de/CG000204_ChromiumNextGEMSingleCell3_v3.1_Rev_D.pdf).
- We added an additional statement in the methods that all samples had at least a 90% viability as determined through trypan blue exclusion [line 111-112]. Although not mentioned in the text, we also observed viability to be in excess of 90% in all samples as determined though 7-AAD staining using flow cytometry (mean = 96.9%; min = 93.2%; max = 99.1).

Critique 2:

The authors should describe how the samples collected and from what site in the OSA, for examples did a pathologist described the degree of necrosis or degree abnormal bone. While the samples were subcategorized in Table 1 was this based on the sample submitted for sequencing or on the H&E section taken by pathology after the fact? OSA is highly heterogeneous and sections can vary in cell type.

Response:

- Due to the timing of sampling and unpredictable cell yield from digestion we did not complete H&E on the same biopsy sample used for scRNA-seq, though separate biopsy samples from the same tumor samples were obtained for and processed for H&E evaluation. If there were sufficient cells were remaining following preparation for single-cell sequencing, we completed flow cytometric analysis on these samples.
- We have updated the methods to provide additional details on sample handling [line 87-92; 96-98; 102-105].

Critique 3:

QC data (web summary) such as estimated number of cells, median reads per cell, reads mapped confidently to the transcriptome were not provided in supplementary data. In addition, the RNA integrity number (RIN) was not reported for the samples.

Response:

- We added an additional supplementary table containing the QC metrics obtained from the 10x Genomics Cell Ranger pipeline. The values in the supplemental data sheet were extracted directly from the 10x genomics html report (Supplemental table 3). In addition, we comment on some of the summary statistics in the first paragraph of the results [line 286-287].
- Please see previous comment regarding the evaluation of the RIN value.

Critique 4:

in the data filtering section, the sentence starting " During this step, we regressed out the percent mitochondrial" provide more information about the term regressed.

Response:

- We added a reference to the manuscript associated with Seurat version 2 where they discuss regression of mitochondrial heterogeneity and cell cycles gene signature scoring prior to running canonical correlation analysis (6). Furthermore we revised our methods to increase transparency of our approach [line 132-133; 137-149].
- The goal of regressing out the percentage of mitochondrial reads in this manuscript was to minimize the impacts of an (usually confounding) aspect cellular metabolic activity on

the integration and clustering of the dataset. Briefly, the approach used the percentage of reads mapping to mitochondrial chromosomes (out of total unique features in a given cell) as latent variable in a linear regression framework applied during scaling of the data.

Critique 5:

In the Cell Classification section cell classification was based on a single marker e.g. CD68 for macrophages etc.

While some markers for canine immune cells are lacking it seems presumptuous to use a single marker to identify cells e.g. while FLT3 is important DC cells is also present on hemopoietic cells.

Response:

- We agree that the use of a single marker to identify a population is inappropriate. Although we initially listed 1 feature for some cell types in the methods, we did not rely on a single marker to identify any cell type within the dataset. Rather multiple algorithmic (singleR, label transfer from human dataset, label transfer from canine PBMC data) and gene set enrichment analysis/enrichment scoring) approaches were used to classify an individual cell type. Our reason for identifying cells with one or a select few features was to provide an easily referenceable feature list for readers viewing the clustering data.
- For feature call outs we expanded on any lists that only included 1 feature. We also provided a supplemental table (Supplemental table 1) with short cell type gene signatures and expanded on the “Cell classification” methods section [line 169-182].

Critique 5:

In the CNV section the authors refer to high quality cells containing 2000 UMIs, how were these cells identified as highly quality without providing QC data.

Response:

- Please see above comments in response to Major Concern 1 from Reviewer 1 in which another question regarding the CNV analysis was raised.
- To specifically address the concern presented here, the 2000 UMI threshold is the default/recommended threshold to use based on the CopyKat protocol (7). The interactive UCSC cell browser will allow for viewing of the UMI (aka nCount_RNA) values for each cell.

Critique 6:

In the human OS homology section, the authors need to explain the sentence "Following annotation of each species, the 12 (6 human and 6 canine) OS samples were integrated into one object to using a SCTransform workflow with 3000 variable features as anchors

(sic)" with reference to integration of data using SCTransform which is used for normalization and variance stabilization of data in the Seurat package.

Response:

- We were inappropriately using the phrase, "SCTransform integration workflow" to refer to the use of integration using canonical correlation analysis of data normalized via SCTransform (Pearson residuals from "regularized negative binomial regression"). We updated the text to be more explicit about what we are referring to. Thank you for bringing this to our attention [line 137-139].

Critique 7:

If the QC data is up to standard then the result drawn by the authors are important. The study is a first in kind in canine OSA. Nevertheless, to verify conclusions drawn from study will require spatial analysis. This because of heterogeneity of the OSA samples and while the study is scRNAseq it comes from a combination of cells i.e. a bulk analysis of the tumor. The authors do acknowledge limitations.

Response:

- Thank you. The QC metrics and the data have been made available in the revised submission. We aim to be as transparent as possible and to make the data (annotated and raw) highly accessible.

Reviewer #3 (summary):

Single-cell RNA sequencing reveals the cellular and molecular heterogeneity of treatment-naïve primary osteosarcoma in dogs.

Ammons et al. here have presented a comprehensive scRNA-seq atlas of canine osteosarcoma. Study represents an important resource for comparative genomics of canine cancers, especially from a point of charting an otherwise elusive canine tumor immune-microenvironment (TMEN). Authors do acknowledge a limited sample size of canine OS samples but that said, all of scRNA-seq analyses are rigorous and well-documented. This study highlights immune cell heterogeneity in terms of cell subtypes and their potential marker genes which could be of value for future immunotherapy trials involving canine osteosarcoma cases. Authors go beyond conventional cell type classification to infer in silico gene regulatory and surfaceome marker activity, the latter being of a potential value - by targeting highly conserved surface tumor antigens - in designing experimental CAR-T therapies in an otherwise unchanged treatment modalities for osteosarcoma patients. Study should be treated as a useful database or resource for cross-species comparison of TMEN and as authors stated, can be of value for future canine cancer studies to deconvolve bulk RNA-seq data using CIBERSORT (which otherwise is not feasible without canine-specific scRNAseq data). I have two comments which - if feasible within timeline and resources - can aid in improving rigor of cell type classification and provide experimental validation of a few key predicted cell types.

Comment 1:

Since core cell type classification relies on UMAP, worth considering an orthogonal approach like CONCORDEX to validate cell type assignments from scRNA-seq data.

Response:

- Thank you for pointing us to CONCORDEX, we explored the use of CONCORDEX and a summary of the results of the analysis is presented below.
- Overall, the analysis provided support that major cell classifications are consistent in the UMAP space as well as through evaluation in the context of K-Nearest Neighbors.
- Specifically, the figures below demonstrate a high degree of confidence in the segregation of cells into the 8 major categories. However, as we increased the resolution of cell type annotations, there was a corresponding increase in uncertainty in the cell type divisions (more shared neighbors between cell types).
- While these data could be interpreted as evidence that the cell types are over clustered, the approach presented in CONCORDEX is meant to function as a supplemental approach to evaluate the performance of clustering and UMAP representation of the data. Our division of cell types relied on the use of subsetting on major cell types, completing independent unsupervised clustering, and careful evaluation of how clusters change as clustering resolution is altered. We evaluated cluster stability by implementing the visual approach presented in the R package clustree to identify “stable” clusters within our dataset (8). An example of the output for the T cell subset is presented below. In this example we determined that the cluster stability was achieved at a resolution of

0.6, which resulted in the identification of the 10 distinct T cell subtypes presented in Figure 3 of the manuscript. While we could have increased the resolution to 0.7 to further divide cluster 0 (corresponds to CD8_{ex} in Figure 3 of manuscript), when we investigated the possible division, we could not define the two clusters as biologically distinct entities (a subjective decision). Similar approaches were applied to analysis and annotation of all cell types.

The number of dims used:40

Comment 2:

A complementary approach to validate a few if not all cell types would be to carry out IHC based on predicted cell markers for scRNA-seq defined cell types, and validate if expression patterns are for such markers is concordant with scRNA-seq dot plots and restricted to a given cell type.

Response:

- In our recent report completing single-cell RNA sequencing on canine leukocytes, we were able to evaluate cell type proportions as determined by flow cytometry and scRNA-seq (9). The analysis revealed a strong correlation between scRNA-seq cell abundances and flow cytometry abundances, suggesting that, in the context of immune cell populations, the two approaches were able to resolve similar major cell type populations. Unfortunately for this manuscript we were unable to obtain adequate cell numbers from individual samples to complete comprehensive flow cytometric analysis. Furthermore, the small sample size further limited the ability to evaluate correlation of cell type proportions. Despite this, the figure provided below depicts the correlation with the limited data we had available.
- Overall, there is moderate correlation between cell type classifications when evaluating the cells as either immune or non-immune (CD45+) by flow cytometry and scRNA-seq.
- While IHC analysis would have been informative, we believe it is beyond the scope of the current work. We also draw the reviewer's attention to the Mannheim et al. 2023 manuscript which completed IHC analysis of canine osteosarcoma biopsies using antibodies to Iba1 (AIF1), CD204 (MSR1), FOXP3, and others to validate some prognostic markers (10). These IHC classifications generally correlated well with the immune cell classifications we identified in the current work.

References

1. Luecken MD, Theis FJ. Current best practices in single-cell RNA-seq analysis: a tutorial. *Mol Syst Biol.* 2019;15(6):e8746.
2. McDermaid A, Monier B, Zhao J, Liu B, Ma Q. Interpretation of differential gene expression results of RNA-seq data: review and integration. *Brief Bioinform.* 2019;20(6):2044–54.
3. Black MB, Andersen ME, Pendse SN, Borghoff SJ, Streicker M, McMullen PD. RNA-sequencing (transcriptomic) data collected in liver and lung of male and female B6C3F1 mice exposed to various dose levels of 4-methylimidazole for 2, 5, or 28 days. *Data Brief.* 2021;38:107420.
4. Squair JW, Gautier M, Kathe C, Anderson MA, James ND, Hutson TH, et al. Confronting false discoveries in single-cell differential expression. *Nat Commun.* 2021;12(1):5692.
5. Tirosh I, Izar B, Prakadan SM, Wadsworth MH, Treacy D, Trombetta JJ, et al. Dissecting the multicellular ecosystem of metastatic melanoma by single-cell RNA-seq. *Science* (1979). 2016;352(6282):189–96.
6. Butler A, Hoffman P, Smibert P, Papalexi E, Satija R. Integrating single-cell transcriptomic data across different conditions, technologies, and species. *Nat Biotechnol.* 2018;36(5):411–20.
7. Gao R, Bai S, Henderson YC, Lin Y, Schalck A, Yan Y, et al. Delineating copy number and clonal substructure in human tumors from single-cell transcriptomes. *Nat Biotechnol.* 2021;39(5):599–608.
8. Zappia L, Oshlack A. Clustering trees: a visualization for evaluating clusterings at multiple resolutions. *Gigascience.* 2018;7(7):giy083.
9. Ammons DT, Harris RA, Hopkins LS, Kurihara J, Weishaar K, Dow S. A single-cell RNA sequencing atlas of circulating leukocytes from healthy and osteosarcoma affected dogs. *Front Immunol.* 2023;14:1162700.
10. Mannheimer JD, Tawa G, Gerhold D, Braisted J, Sayers CM, McEachron TA, et al. Transcriptional profiling of canine osteosarcoma identifies prognostic gene expression signatures with translational value for humans. *Commun Biol.* 2023;6(1):856.

REVIEWERS' COMMENTS:

Reviewer #1 (Remarks to the Author):

The authors have addressed my concerns thoroughly. I particularly appreciate their efforts to make data and analysis code available. Please do make sure these are released publicly at the appropriate time as they will be a useful resource to the community.

Reviewer #3 (Remarks to the Author):

Thanks for working on revisions, including addition of CONCORDEX and complementary approaches to ensure statistical rigor. Also, appreciate updating SRA repository. Best regards.